# Thin ice clouds in the Arctic: Cloud optical depth and particle size retrieved from ground-based thermal infrared radiometry

Yann Blanchard[1], Alain Royer[1], Norman T. O'Neill[1], David D. Turner[2], and Edwin W. Eloranta[3]

[1]Centre d'Applications et de Recherches en Télédétection, Université de Sherbrooke, Sherbrooke, Québec, Canada.
[2]Global Systems Division, Earth System Research Laboratory, National Oceanic and Atmospheric Administration, Boulder, Colorado, USA.
[3]Space Science and Engineering Center, University of Wisconsin, Madison, Wisconsin, USA.

*Correspondence to:* Yann Blanchard (yann.blanchard@usherbrooke.ca)

**Abstract.** Multi-band downwelling thermal measurements of zenith sky radiance, along with cloud boundary heights, were used in a retrieval algorithm to estimate cloud optical depth and effective particle diameter of thin ice clouds in the Canadian high-Arctic. Ground-based thermal infrared (IR) radiances for 150 semi-transparent ice clouds cases were acquired at the Polar Environment Atmospheric Research Laboratory (PEARL) in Eureka, Nunavut, Canada (80°N, 86°W). We analyzed and quantified the sensitivity of downwelling thermal radiance to several cloud parameters including optical depth, effective particle diameter and shape, water vapor content, cloud geometric thickness, and cloud base altitude. A look up table retrieval method was used to successfully extract, through an optimal estimation method, cloud optical depth up to a maximum value of 2.6 and to separate thin ice clouds into two classes: 1) TIC1 clouds characterized by small crystals (effective particle diameter $\leq$ 30 μm), and 2) TIC2 clouds characterized by large ice crystals (effective particle diameter > 30 μm). The retrieval technique was validated using data from the Arctic High Spectral Resolution Lidar (AHSRL) and Millimeter Wave Cloud Radar (MMCR). Inversions were performed across three polar winters and results showed a significant correlation ($R^2$ = 0.95) for cloud optical depth retrievals and an overall accuracy of 83% for the classification of TIC1 and TIC2 clouds. A partial validation relative to an algorithm based on high spectral resolution downwelling IR radiance measurements between 8 and 21 μm was also performed. It confirms the robustness of the optical depth retrieval and the fact that the broadband thermal radiometer retrieval was sensitive to small particle (TIC1) sizes.

## 1   Introduction

Predictions of future climate change and its regional and global impacts require that a better understanding of the radiative transfer interactions between clouds, water vapor and precipitation be incorporated into appropriate models. Recent CMIP5 model intercomparisons (the Coupled Model Intercomparison Project as described in Jiang et al., 2012) indicate large variability in ice cloud parameters (for example ice water content) amongst high-latitude models. Shortcomings in ice cloud parametrization (Baran, 2012) impact their representation of radiative effects as well as water cycles and leads to uncertainties in quantifying cloud feedbacks in the context of climate change (Waliser et al., 2009). High-altitude thin ice clouds consisting of pure ice crystals, which cover between 20 to 40% of the Earth (Wylie and Menzel, 1999), can, for example, have opposing

effects on the radiative properties of the Earth. A surface cooling effect ensues when scattering by clouds reduces the solar radiation reaching the Earth's surface (i.e., albedo effect). By contrast, a reduction in the amount of IR energy escaping the Earth-atmosphere system occurs when the upwelling IR radiation emitted by the Earth's surface and lower atmosphere is absorbed by clouds and radiated back downward (i.e., greenhouse effect) (Stephens and Webster, 1981). The macrophysical and microphysical properties of thin ice clouds determine which process dominates and hence determine the net forcing of thin ice clouds on the climate system (Stephens, 2005). The optical properties of thin ice clouds can be represented by extensive parameters such as the optical depth and ice water content as well as intensive parameters such as ice crystal size and shape. In an Arctic environment, the radiative effects of ice clouds are unique because their radiative forcing influence on the energy balance depends on seasonal Polar day to Polar night variation as well as large scale processes like the Arctic Oscillation (Wang and Key, 2003). The advent of active sensors onboard satellites (for example CALIPSO/CloudSat) has enabled the application of considerably more resources for polar region ice cloud studies. This permits the evaluation of climate models (Jiang et al., 2012) and satellite cloud climatologies (Sassen et al., 2008). Long-term ground-based observations which are also essential for the validation of models and satellite climatology are, however, limited in their Arctic coverage (Heymsfield et al., 2017).

Thermal IR radiometry is a well-known technique for investigating the presence and the emissivity of clouds (Allen, 1971). Numerous researchers have exploited the thermal IR behavior of the absorption and scattering efficiencies of cloud particles as a means of retrieving CODs and particle effective sizes (e.g., Inoue, 1985). As cloud altitudes (temperatures) can lead to large uncertainties in this latter technique, Platt (1973) proposed using lidar backscatter profiles along with IR radiometry to estimate cloud altitudes and accordingly improve the retrieval accuracy of cloud emissivity. This active/passive technique (called LIRAD for lidar/radiometer method by Platt) has evolved over the years with such improvements as the availability of high resolution spectrometers (Smith et al., 1993; Lubin, 1994). The LIRAD technique is based on spectral radiance/brightness temperature comparisons between measurements and radiative transfer calculations. It performs better in the presence of high thermal contrast and is thus well suited for cloud retrievals (Lubin, 1994). In more recent applications, cloud optical depth, effective radius and ice fraction were retrieved from AERI (Atmospheric Emitted Radiance Interferometer; Knuteson et al., 2004a, b) spectral downwelling radiance observations in Antarctic, during the Surface Heat Budget of the Arctic Ocean (SHEBA) campaign and ARM North Slope of Alaska site (Turner et al., 2003; Shupe et al., 2015; Mahesh et al., 2001, respectively). The Turner (2005) method was also employed at Eureka (Nunavut, Canada) to retrieve cloud optical depth and cloud microphysical parameters from AERI spectra acquired between 2006 and 2009 (Cox et al., 2014). A proposed satellite-based instrument whose goal will be the characterization of thin ice clouds in the Arctic using far and thermal infrared channels (Blanchet et al., 2011) was recently tested during an airborne campaign in the High Arctic (Libois et al., 2016).

In this paper, we examine how multi-band thermal measurements of zenith sky radiance can be used to retrieve what are, as indicated in the early remote sensing literature (see Nakajima and King, 1990, for example), the most critical extensive and intensive parameters influencing the radiative effects of ice clouds: cloud optical depth (COD) and effective particle diameter ($D_{eff}$). We propose an application of this LIRAD technique with a relatively simple and inexpensive instrument (less than 10% of the cost of an AERI) that is well-suited to the Arctic environment (Royer et al., 2014). Inasmuch as ice particle size is difficult to retrieve from IR radiometry, the $D_{eff}$ component of our retrievals will be focused on a simple discrimination of

large and small crystal sizes. This approach was motivated by previously published research that indicated such a discrimination would play a key role in characterizing an important aerosol/cloud interaction process in Polar winter, namely precipitative cooling (see, for example, Blanchet and Girard, 1994; Grenier et al., 2009). An important aspect in this paper is that our COD and $D_{eff}$ retrievals will be validated using independent lidar and radar retrievals.

Section 2 highlights the importance of studying ice clouds while Section 3 is devoted to the description of the study site and instrumentation. Section 4 examines the sensitivity of thermal IR radiometry to key ice cloud parameters. In Section 5 we describe and verify the proposed methodology for retrieving COD and $D_{eff}$ using thermal IR radiometry measurements. The results are presented and discussed in Section 6 for the 150 thin ice clouds cases we observed in the Arctic.

## 2 Classification and Parameterization of Thin Ice Clouds

Water vapor and clouds are a significant climate modeling challenge since they represent major radiative forcing influences, while being the least understood components of the climate system (Waliser et al., 2009; Jiang et al., 2012). Much of the recent research has been focused on aerosol-cloud interactive processes involving aerosols acting as ice and water cloud nuclei and their subsequent affect on cloud microphysics, precipitation and radiation (see for example, Feingold and McComiskey

(2016) on recent ARM campaigns, Winker et al. (2010) and Illingworth et al. (2015) respectively, on the cloud remote sensing mandate of the A-Train and EarthCARE satellite missions and Jouan et al. (2014) as part of the NETCARE project). In particular, understanding aerosols and their radiative effects, especially their indirect impacts as cloud condensation nuclei, is of critical importance for climate change models. The indirect effect of aerosols represents a cooling influence (whose amplitude is subject to large uncertainties) on the global radiative budget (Intergovernmental Panel on Climate Change, IPCC, 2013). The

estimated uncertainty of the indirect forcing component (between -0.1 and -1.3 $W.m^{-2}$) is associated with variations in cloud properties and cloud lifetime. In the Arctic, the nature of thin ice clouds can effectively induce an indirect cooling influence given the proper conditions. In terms of the purpose and motivation for this paper, we note that the presence of sulphuric-acid bearing aerosols (viz., Arctic haze) can significantly increase the size of ice particles (relative to the size of ice particles formed from more pristine, low acid aerosols or supercooled droplets). This process can cause enhanced precipitation and important

cooling effects during the polar winter and could possibly lead to a dehydration greenhouse feedback (DGF) effect, as proposed by Blanchet and Girard (1994).

The small and large ice particles described above are often abbreviated as TIC1 and TIC2 (thin ice cloud, type 1 and 2). Thin ice cloud classification was carried out by Grenier et al. (2009) using the active techniques of lidar and radar: CALIPSO and CloudSat data were employed to discriminate between TIC1 and TIC2 ice clouds using the CloudSat small-particle sensitivity

minimum of approximately 30-40 μm. In this study, we seek to demonstrate that TIC1 and TIC2 discrimination can be determined using zenith-looking IR radiance measurements acquired at the Eureka observatory in the Canadian High Arctic. Figure 1 illustrates lidar and radar backscatter profiles acquired at Eureka for distinct TIC1 and TIC2 cases.

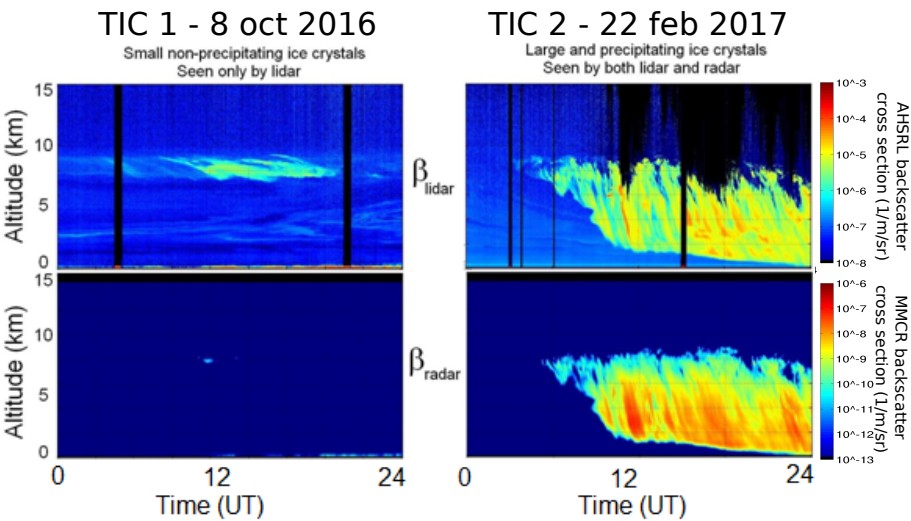

**Figure 1.** Classification of thin ice clouds using ground-based lidar (up) and radar backscatter profiles (bottom).

The left hand lidar profile of Figure 1 shows a TIC that is largely transparent to cloud radar while the right hand lidar profile shows a thin ice cloud that is readily detected by the radar. The disparity in radar detectivity enables one to conclude that the former case corresponds to a small-particle TIC1 event while the latter case corresponds to a large-particle TIC2 event.

The presence or absence of thin ice clouds in the winter can lead to significant changes in surface cooling (Stephens et al.,

1990). Given the important radiative influence of ice crystal size and COD it is necessary that these parameters be well-characterized in order to improve modeling of their radiative effects (Ebert and Curry, 1992) and thus their influence within a context of TIC1 and TIC2 clouds.

The effective diameter of atmospheric ice particles is defined by Hansen and Travis (1974):

$$D_{eff} = \frac{3V}{2A} \tag{1}$$

where A and V are respectively the total projected area and volume of all ice particles per unit surface area in a given atmospheric column (Baum et al., 2014). COD is given by:

$$COD = \int \pi Q_{ext} \frac{D^2}{4} a(D) dD \tag{2}$$

where $Q_{ext}$ is the extinction efficiency (extinction cross section per unit projected-particle-area) (Hansen and Travis, 1974), D is the particle diameter and $a(D)$ is the ice particle number density per unit increment in diameter. We note that, while the

lidar-derived CODs employed in this article are at 0.532 μm, the IR CODs from our retrieval method were referenced, for convenience, to 0.55 μm (we assume that COD differences between 0.532 and 0.55 μm are negligible within the context of other uncertainties encountered in this study).

**Table 1.** List of the Eureka (PEARL) instruments employed in our analysis.

| Ground-based instrument | Duty cycle | Research group or institution |
|---|---|---|
| AHSRL (Arctic High Spectral Resolution Lidar) | Continuous | SEARCH/NOAA - U. of Wisconsin |
| MMCR (Millimeter Cloud Radar) | Continuous | SEARCH/NOAA - ARM |
| Radiosonde | Twice a day | Environment Canada |
| P-AERI (Polar-AERI) | Continuous (except during precipitation) | SEARCH/NOAA - U. of Idaho |

## 3   Study site and instrumentation

The observation site was the Polar Environment Atmospheric Research Laboratory (PEARL) in Eureka, Nunavut (80°N, 86°W) which is one of the high-latitude stations of the Network for the Detection of Atmospheric Composition Change (NDACC, http://www.ndsc.ncep.noaa.gov/sites/stat_reps/eureka/). This high Arctic site is located in the northernmost part of the Canadian Arctic Archipelago. It was chosen because of our interest in Arctic ice clouds and to exploit the diverse and complementary inventory of atmospheric instruments listed in Table 1 (i.e., lidar, radar, IR spectrometer and radiosondes), as well as the infrastructure and logistics support for field campaigns.

Detailed descriptions of the AHSRL and MMCR data processing and interpretative techniques can be found in Eloranta (2005) and Moran et al. (1998). A summary of instrument specifications is given in Bourdages et al. (2009). Knuteson et al. (2004a, b) present a discussion of the AERI performance which is applicable to the present paper. The AERI instrument is known to have a very small warm bias for low radiance measurements, typically for clear-sky events, on the order of 1% of the ambient radiance (Knuteson et al., 2004b; Delamere et al., 2010). As the focus of our work is on clouds with COD greater than 0.1, this warm bias in the AERI has only a slight to negligible impact for retrievals involving very thin clouds (Turner, 2003). The AERI data used in this work have been post-processed to reduce the uncorrelated random error in the data using principal component analysis (Turner et al., 2006).

In this paper, we focus on the potential of using data from a ground-based multi-band thermal radiometer, the CIMEL CE-312 developed by CIMEL Inc (see Legrand et al., 2000; Brogniez et al., 2003, for descriptions of a similar instrument). The 6 channels of this radiometer correspond to (full width at half maximum) limits of 8.2-8.6, 8.5-8.9, 8.9-9.3, 10.2-10.9, 10.9-11.7 and 11.8-13.2 μm and filter response peak values at 8.4, 8.7, 9.2, 10.7, 11.3 and 12.7 μm. The multi-band radiometer is also a robust instrument, that, unlike the AERI, does not require a thermally controlled environment. In actual fact however, we had to simulate the response of this radiometer by convolving the spectral transmittance of each filter with the spectra of the Eureka Polar AERI (P-AERI) instrument (provided by Von Walden at the U. of Idaho and NOAA). The reason for this was that the CIMEL radiometer that we hoped to use was not ready for deployment when we performed the field campaigns.

## 4  Sensitivity of Thermal Infrared Radiometry to Thin Ice Clouds

The spectral sensitivity of longwave (thermal) radiation to the microphysical properties of ice clouds has been investigated for satellite data (Chiriaco et al., 2004; Dubuisson et al., 2008), airborne data (Brogniez et al., 2004; Libois et al., 2016) and ground-based sensors (e.g. Comstock and Sassen, 2001; Yang et al., 2005, and articles cited above). Previous studies have
demonstrated that thermal IR radiometry is relevant in terms of permitting the retrieval of both COD and, to a degree, $D_{eff}$. The retrieval of the latter parameter permits, in turn, a discrimination of TIC1 and TIC2 clouds. The dependence of thermal IR radiometry on ice particle size is represented by equation (2). The extinction efficiency, a measure of particle attenuation (absorption and scattering), depends on particle size, composition and shape as well as wavelength (Hansen and Travis, 1974). It is common, in the case of zenith-looking thermal IR (8-14 μm) radiometry of ice clouds, to neglect the scattering portion of
$Q_{ext}$ (where $Q_{ext} = Q_{abs} + Q_{sca}$), especially for large particles (Platt, 1973). The result in the presence of a medium such as cloud is extremely simplified radiative transfer that is characterized by a strong forward-scattering phase function: in the limit of a delta-function phase function, all forward scattered radiation in any given direction of incidence is returned to the incident beam and the only radiance loss is due to absorption (see, for example, the delta-function irradiance solution of Meador and Weaver, 1980). Platt (1973) and later authors such as (Turner and Löhnert, 2014) indicated that only a small fraction of the
zenith-looking downwelling radiation emitted by a cloud was due to scattering (in spite of the fact that $Q_{sca} \approx Q_{abs}$).

We accordingly chose to plot absorption efficiency spectra in Figure 2 in order to illustrate the spectral sensitivity of this key radiative transfer parameter. The absorption efficiency of TIC1 particles ($D_{eff}$ from 10 to 30 μm) and TIC2 particles ($D_{eff}$ from 35 to 120 μm) for "severely roughened solid column" type crystals (Figure 2a and 2b) were obtained from calculations reported by Yang et al. (2013) and Baum et al. (2014). These spectra were then replaced by their mean and standard deviation
across the two (TIC1 and TIC2) $D_{eff}$ regimes in order to better appreciate the band to band separability of the TIC1 and TIC2 size classes (Figure 2c). Prior to computing those means and standard deviations of Figure 2c, we integrated the individual spectra of Figures 2a and 2b across the pass-bands of the six channels employed in this study (triangles in Figure 2c). The $D_{eff}$ ranges employed to define TIC1/TIC2 particles, for the averaging carried out in the creation of Figure 2c, were, as in Grenier et al. (2009), roughly based on their non-detectability to detectability threshold in radar backscatter returns.

This coarse spectral representation obtained for the 6 band averages and standard deviations enables one to better appreciate the more robust nuances between the two families of spectral curves (especially for the 8.4, 8.7, 9.2 and 12.7 μm channels) and better understand the key discriminatory elements of the classification into TIC1 and TIC2 clouds. It is clear from Figure 2c that the first 4 bands offer the greatest potential for discriminating particle size. Figures 2a and 2b however indicate a decreasing sensitivity to increasing particle size as one approaches $D_{eff}$ values in the tens of μm.

We simulated the influence of COD and $D_{eff}$ variations, on brightness temperature ($T_b$) variations, using the MODTRAN 4 radiative transfer model (Berk et al., 1999). Figure 3 shows simulated $T_b$ variations for the six radiometer channels as a function of COD for fixed $D_{eff}$ and as a function of $D_{eff}$ for fixed COD. The fixed values of $D_{eff}$ and COD (and other independent parameters of the MODTRAN 4 runs) correspond to a reference case whose parameters are defined in Table 2. We chose the input parameters of the reference case as the set of mean parameters obtained by averaging over the parameters

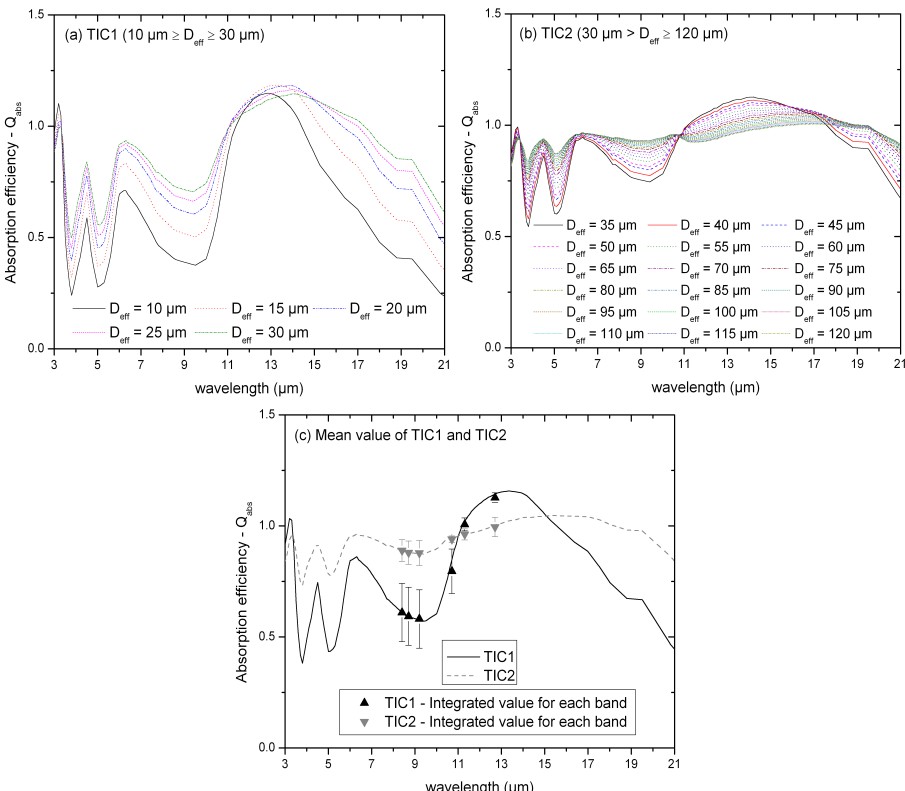

**Figure 2.** 2a and 2b - Absorption efficiency spectra for TIC1 and TIC2 particles across a range of $D_{eff}$ values, 2c - Mean absorption efficiency and standard deviations across the spectra of Figures 2a and 2b. The triangular symbols represent the integration of the absorption efficiency across the six bands of the CIMEL CE-312. These efficiencies were derived for the "severely roughened solid column" type crystals of Yang et al. (2013) and Baum et al. (2014), available at: http://www.ssec.wisc.edu/ice_models/polarization.html

of the 150 cloud cases that we employed to provide an empirical validation of our retrieval (see Section 6.2 and Table 2 for more details). The curves of Figure 3 represent an illustrative subset of our inversion lookup table (LUT) that we employed as a means of retrieving COD and $D_{eff}$ from measured values of $T_b$. The $D_{eff}$ column is biased by the fact that the lidar-radar retrieval is insensitive to TIC1 particles: the $D_{eff}$ reference value was accordingly biased downward to roughly overcome this

5  insensitivity. The value of 50 µm was also the value employed by Sourdeval et al. (2013).

Figure 3a indicates that, at a fixed $D_{eff}$ value of 50 µm, there is a strong and monotonic variation in $T_b$ as a function of COD for all channels. At COD magnitudes greater than 2-3, the $T_b$ values for all channels converge towards an asymptotic ceiling that is the brightness temperature of an opaque representation of the cloud. This clearly shows that the sensitivity of the method decreases progressively as the COD increases beyond 3.

10  Differences in $T_b$ behavior over a range of $D_{eff}$ values and a fixed COD of 0.5 can be observed in Figure 3b. For the channels of nominal wavelength less than 10.7 µm, $T_b$ varies monotonically with $D_{eff}$ up to approximately 30 µm, after which

**Table 2.** Average, standard deviation, maximum and minimum values of the parameters of the cloud cases used in this study. The reference case, defined in the last row, was employed to produce Figure 3 (while varying $D_{eff}$ and the COD) and was used for the sensitivity study of Figure 4. The means, standard deviations and extrema of each parameter were derived from our analysis of the 150 cloud cases. For the reference case, the cloud base height and thickness values were, for the sake of convenience, rounded to the nearest incremental step of the MODTRAN vertical layer profiles.

| | Cloud base height (km) | Thickness (km) | Water vapor content ($g/cm^2$) | COD (lidar) | Deff (μm) (lidar-radar) | Ice particle shape (3 sets of models) |
|---|---|---|---|---|---|---|
| Average | 5.19 | 2.30 | 0.19 | 0.46 | 91.06 | -Solid columns |
| Std Dev | 2.05 | 1.62 | 0.09 | 0.48 | 25.16 | -Aggregate of solid columns |
| Max | 9.00 | 8.00 | 0.85 | 2.60 | 158.26 | -A mixture involving a set |
| Min | 0.80 | 0.20 | 0.08 | 0.10 | 40.07 | of 9 habits |
| Reference case | 5.20 | 2.20 | 0.19 | 0.50 | 50.00 | Solid columns |

A full description of the models can be found at http://www.ssec.wisc.edu/ice_models/polarization.html

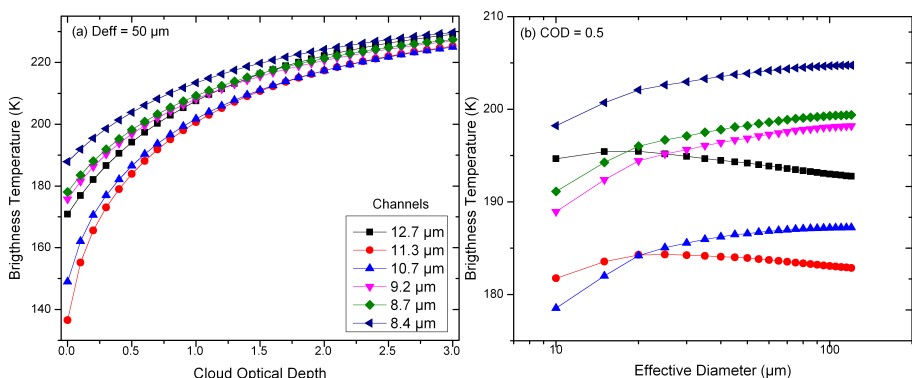

**Figure 3.** Variation in brightness temperature ($T_b$) with (a) cloud optical depth (COD) and (b) effective diameter ($D_{eff}$) for the six bands of the CIMEL CE-312. The color legend of the left hand graph applies to both graphs. The MODTRAN input parameters for this reference case are detailed in Table 2.

the response plateaus to variations of $\approx$ 1K or less. For the 11.3 and 12.7 μm channels, the responses are non monotonic (or considerably less monotonic) for the smaller values of $D_{eff}$ and smoothly decrease with increasing $D_{eff}$ beyond a peak in the 10 - 20 μm range. This decrease is associated with the relatively large spectral changes seen in the refractive index of ice particles at these larger wavelengths (see for example Warren and Brandt, 2008).

5   As discussed above, IR radiance measurements are sensitive to a variety of cloud parameters as well as to the cloud environment. The simulation results in Figure 4 detail the band dependent effects of six different parameters by comparing changes in $T_b$ induced by each parameter individually. These were obtained for 1000 appropriately normalized samples of a random number generator with a normal probability distribution whose mean and standard deviation was controlled by the six parameter

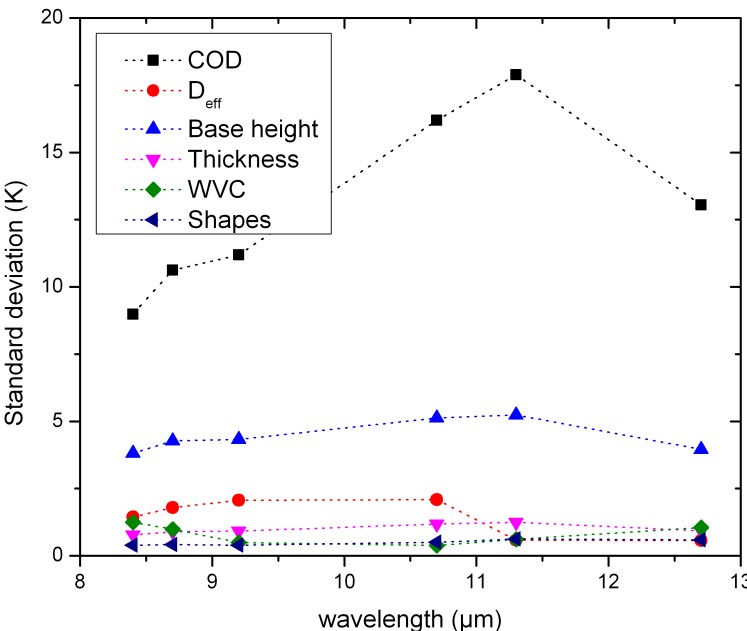

**Figure 4.** Sensitivity of $T_b$ as a function of six key radiative transfer parameters. The standard deviations (in units of K) are obtained by stochastically varying, with a sample size of 1000, the parameters of interest within the limits given in the Table 2.

values of Table 2 (COD, $D_{eff}$, cloud base height, cloud thickness, column integrated water vapor of the atmosphere (WVC) and particle shape). The particle shape parameter is based on three particle characterizations as defined by Baum et al. (2014): severely roughened solid columns, a general habit mixture involving a set of 9 habits, and severely roughened aggregate of solid columns. The standard deviations in $T_b$ that result from the variation of the six parameters are computed relative to the

reference case defined above (Table 2). We note that there was little sensitivity to the choice of a 50 μm effective diameter for the reference case: changing this typical TIC2 value to a value more representative of TIC1 particles produced differences of less than 1 K in Figure 4 .

Figure 4 shows that the chosen COD variation had the strongest $T_b$ influence of all of the parameters, especially for the bands at 10.7 and 11.3 μm (as one could infer by referring to Figure 3a). Changes in $D_{eff}$ (in red) lead to a standard deviation around

2 K for the four first bands as can be qualitatively appreciated by referring to Figure 3b. Changes in the altitude of the cloud induce a standard deviation up to 5 K. Indeed, because measurements of thermal IR radiometry are sensitive to temperature, a change in altitude causes a $T_b$ difference that is sensitive to the range of temperatures within which the cloud is located. Cloud thickness and WVC are marginally important parameters in terms of the magnitude of the changes induced in $T_b$. If the altitude and the thickness of the clouds are known from vertical lidar (and radar) profiles then the magnitude of the altitude and cloud

thickness uncertainties of Figure 4 will fall to levels commensurate with the standard deviations ascribed to the uncertainty in the ice particle shape ($\approx$ 1 K as per the dark blue curve). This latter uncertainty (determined from the habit parameterizations listed in Table 2) is largely inflexible inasmuch as our ability to distinguish ice particle shape from lidar depolarization data

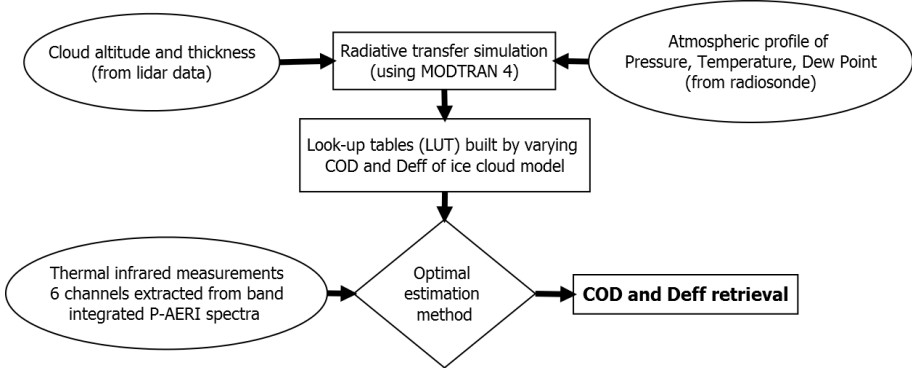

**Figure 5.** Flowchart of the retrieval method.

is extremely limited. Water vapor content (WVC) in the atmosphere, which remains relatively low during the polar winter at Eureka, has a weak absorption influence on the radiance measurements acquired in the CE-312 bands. Its associated uncertainty is commensurate with the uncertainties due to particle shape and cloud thickness: however integrated WVC is estimated from radiosonde profiles at Eureka and thus its uncertainty can be reduced to levels significantly below the particle shape and

cloud thickness uncertainties. These reductions in the uncertainty of nominally known input parameters will be such that the variability of the parameters to be inverted (COD and $D_{eff}$) is significantly larger than the uncertainty of the known input parameters (for all bands in the case of the COD and at least in the case of the first 4 bands for $D_{eff}$).

## 5   Methodology

Our LIRAD objective was performed using LUTs and MODTRAN 4 radiative transfer simulations to parameterize the behavior

of the downwelling zenith sky radiance as a function of key input parameters, including COD and $D_{eff}$. The methodology is represented in the flowchart of Figure 5. The core of this method, inspired by ground-based retrievals (e.g. Turner, 2005), is to compare thermal IR radiance measurements with LUTs derived from MODTRAN 4 simulations.This inverse problem is solved using the optimal estimation method (OEM) (Rodgers, 2000). The method seeks the state of maximal probability, conditional on the value of the measurements, associated errors and a priori knowledge. This OEM is an efficient inversion

method that has already been employed for ice cloud retrievals (see for example Sourdeval et al., 2013).

The steps of the retrieval method are as follows:

1. First, knowledge of the cloud environment at the time of a given radiometer measurement is required. Specific input auxiliary data includes pressure, temperature and water vapor profiles from radiosonde data and the effective cloud-layer height from lidar backscatter data. The radiosonde parameters are interpolated to the radiometer times while the time

of the selected lidar profile is the nearest to the radiometer time. To avoid the issue of interpolating radiosondes over extensively long periods of time, the cases were selected as close as possible to radiosonde launch times. Radiosonde humidity sensors are known to be subject to dry bias especially in dry conditions and could yield relative humidity

underestimates of 10 % (Rowe et al., 2008). The 6 channels are however far less sensitive to WVC than to COD (see Figure 4) and therefore the bias is expected to be lower in cloudy conditions. Cloud heights are estimated for sustained cloud features where clouds are defined by lidar backscatter coefficients greater than $1.10^{-6}$ m$^{-1}$.sr$^{-1}$ and a lidar depolarization ratio greater than 20 % (thresholds were inspired by Shupe (2007) but adapted to a different vertical resolution of our lidar). Upper and lower cloud boundaries are then obtained where 4 continuous vertical samples of the lidar profile ($4 \times 30 = 120$ m for an AHSRL resolution of 30 m) comply with that requirement (preceded by a series of lower, non-cloud, samples).

2. Using MODTRAN 4, we simulated surface based zenith-looking brightness temperatures of a cloudy atmosphere as a function of the environmental data. A LUT is then constructed for 23 values of $D_{eff}$ between 10 and 120 μm and 31 values of COD (from 0 to 3 with an increment of 0.1). Because $T_b$ is so strongly dependent on COD, the LUT is linearly interpolated between MODTRAN 4 calculations with a COD increment of 0.01.

3. Brightness temperatures are then derived from radiance measurements in the six CIMEL CE-312 radiometer channels extracted from band integrated P-AERI spectra.

4. The OEM was used to compare the LUT spectra with the measured $T_b$ spectra. This method requires precise quantification of errors attributed to each variable of the state and measurement vectors (as detailed in Rodgers, 2000, and in Appendix A). We retrieve the best estimates of COD and $D_{eff}$ from the most optimal fit to the measured $T_b$.

Specific validation elements for our retrieval algorithm included profiles of the effective ice particle diameter prime ($D'_{eff}$) that were extracted from the combination of AHSRL and MMCR backscatter coefficients. This was carried out as per the technique developed by Donovan and van Lammeren (2001) and applied to the instruments at our study site (Eloranta et al., 2007). $D'_{eff}$ is given by:

$$D'_{eff} = \sqrt[4]{\frac{\beta_{radar}}{\beta_{lidar}}} \tag{3}$$

where $\beta_{radar}$ and $\beta_{lidar}$ are the extinction cross-sections of the radar and lidar respectively. $D'_{eff}$ can be related to $D_{eff}$ assuming an analytical form for the size distribution which in our case was taken as a modified gamma distribution of hexagonal columns (Eloranta et al., 2007). In order to compare $D_{eff}$ with our retrievals, we averaged this parameter over the vertical extent of a given cloud . We chose, for simplicity's sake, to assume a specific particle shape (i.e. the hexagonal column shape) when retrieving $D_{eff}$ from the lidar/radar profiles to enable consistent comparisons with our passive retrievals. As a general quality assurance step for the radar data, those cases for which the radar backscatter coefficient was less than $10^{-15}$ m$^{-1}$.sr$^{-1}$ (an empirically determined value of minimum detectability) were eliminated from any retrieval processing (this generally meant the elimination of TIC1 points). It is also important to state that radar signal is proportional to the sixth power of the hydrometeor diameter, whereas IR instruments are sensitive to the ratio of the third to the second moment. This means that the equivalent $D_{eff}$ is not strictly the same and their comparison can generate biases in some conditions (see discussion in Turner, 2005).

The CODs from the passive algorithm was validated by comparison with estimates of COD derived from AHSRL observations using equation (4) and averaged over the cloud geometric thickness. By its design, the AHSRL measures two signals which can be processed to yield separate lidar returns for aerosol and molecular scattering, and then to make reliable measurements of the extinction profile. The optical depth $\tau$ across a range interval $(r, r_0)$ is computed as:

$$5 \quad \tau(r) - \tau(r_0) = \frac{1}{2}\ln\left(\frac{\rho(r)}{\rho(r_0)}\right) - \frac{1}{2}\ln\left(\frac{S_m(r)}{S_m(r_0)}\right) \tag{4}$$

where $\rho$ is the molecular density, and $S_m$ is the range-squared, background corrected, molecular lidar return of the AHSRL. COD was calculated, using equation (4), across the cloud layer where the vertical cloud boundaries were determined according to the backscatter coefficient and depolarization criteria described above. Due to the very small angular field-of-view of the AHSRL receiver (45 μrad), it is common to assume that the backscatter return is negligibly affected by multiple scattered

photons (Eloranta et al., 2007). However, to better quantify the effect of multiple scattering, we applied Eloranta's (1998) approximate model to calculate the multiply scattered lidar returns for the 150 cloud cases. The inputs included molecular and particulate backscatter coefficients, effective diameters (inferred from the AHSRL+MMCR technique discussed below) and cloud height boundaries. The approximate model was employed to compute multiple scattering returns up to the 4th order. The impact of multiple scattering (MS) can then be evaluated in terms of COD inasmuch as multiple scattering lowers the apparent

COD. From Equation (4), we define $\Delta COD_{ms}$ as:

$$\Delta COD_{ms} = -0.5\log(1.0 + Pt/P1) \tag{5}$$

where $Pt$ is an output of the MS model, representing all orders of multiple scattering while $P1$ is the single scattering return. This term can be used to correct the retrieved optical depth by inserting a multiplicative factor $\eta$, as per Platt (1973) to correct for the reduction of the extinction coefficient. The parameter $\eta$ is not constant (Bissonnette, 2005) and Platt argued that it

should vary between 0.5 and 1. Our best estimate of $\eta$ over the ensemble of test cases was 0.95. This factor was used to correct the lidar-derived CODs that we employed for validation purposes in this manuscript.

Although intervals of P-AERI spectra (wavelength range of 3 - 20 μm) were used to simulate the response of the CIMEL radiometer bands, the P-AERI instrument has more extensive capabilities and is sensitive to a larger range of $D_{eff}$, according to the absorption efficiency spectra (see Figure 2 in this article or Figure 5 in Yang et al., 2003). The mixed-phase cloud retrieval

algorithm (MIXCRA) (Turner, 2005) is designed to estimate microphysical properties of both the ice and liquid components of a cloud using spectral IR radiances supplemented with data from various instruments. By using the spectral behavior of several "microwindows" between gaseous absorption lines in the thermal and far IR, MIXCRA can determine cloud phase and retrieve COD and $D_{eff}$ (a detailed description of the algorithm can be found in Turner (2005)). Turner and Eloranta (2008) have demonstrated good agreement between the MIXCRA retrievals and HSRL optical depth measurements during an

experiment at the ARM NSA site. Cox et al. (2014) describe the specifics of the Eureka implementation, including the auxiliary measurements that were employed (notably the AHSRL, MMCR, radiosonde data and a microwave radiometer). Within the scope of this current study, the cloud-phase determination from MIXCRA is used to ensure the comparison of ice-only cloud properties (i.e. those MIXCRA retrievals that yielded negligible liquid water path, LWP < 0.2 g.m$^{-2}$, were taken as being pure ice-cloud cases). MIXCRA results are used here as an alternative point of reference for our retrievals.

## 6 Results and Discussion

### 6.1 Physical Coherence of a Specific Case Study

In this section we seek to illustrate the temporal variation of particle size and COD in a precipitating cloud and demonstrate that our retrieval gives physically coherent results for a specific case that was chosen to exercise both the COD and $D_{eff}$ retrievals. Figure 6 shows the selected 2009 winter campaign case where we compare AHSRL backscatter coefficient, the MMCR backscatter coefficient profile, the $D_{eff}$ profile (which, as pointed out above, is related to $D'_{eff}$) and the results of our inversion (Figures 6a, 6b, 6c and 6d respectively).

Radar reflectivity is commonly used to describe the reflection, scattering and diffraction effects of a target on the incident signal. Radar reflectivity, expressed in $\mathrm{dBZ}$, is logarithmically proportional to the backscatter coefficient and is proportional to the sixth power of the hydrometeors diameter (Battan, 1973). However, to ensure a consistent approach within the context of $D_{eff}$ retrieval, we chose to display $\beta_{radar}$ in Figure 6b.

One can see (Fig. 6d) that $D_{eff}$, for the lidar-radar technique (in blue), increases from 44 µm to 103 µm. This increase appears, in turn, to be correlated with cloud precipitation as evidenced by the accompanying decrease in altitude of the cloud structure seen in the lidar and radar profiles as well as increasing values of the radar Doppler fall velocity profiles (not shown). The passive $D_{eff}$ retrievals (the green colored curve of Figure 6d) show a roughly similar trend from 20:00 to 23:00 (largely characterized however, by significantly smaller $D_{eff}$ values). The insensitivity of the latter retrieval to larger size particles during the period from about 19:00 to 22:30 and the sudden jump in retrieved $D_{eff}$ value after that time is the result of the type of asymptotic ceiling that one sees in Figure 3b and the choices made in the LUT algorithm retrieval: as one approaches the asymptotic ceiling from smaller $D_{eff}$ values, there is clearly a progressive increase in the range of acceptable $D_{eff}$ values for a given $\Delta T_b$ (a decrease in the robustness of the retrieved value). This example also illustrates an important issue related to our TIC1/TIC2 classification goal, where some points, around 20:00 appear to be classified as TIC1 particles by our algorithm while the lidar-radar values between 60 and 80 µm would be classified as TIC2 particles. One possible explanation is that the cloud vertical inhomogeneity, as evident in the cloud structure observable in Fig. 6a is the source of the misclassification. The regions of the effective diameter that are not detected by the lidar-radar retrieval (see the profiles of Fig. 6c) are indeed optically thick regions having more impact on the radiometric retrieval of Fig. 6d and that likely contain smaller particles.

The COD retrievals are, as one would expect, visually coherent with the general strength and extent of the lidar backscatter coefficient. After 2300 UTC our COD retrievals approach the limit of retrieval sensitivity suggested in Figure 3. This is manifested by an artificial non-monotonic increase in the variability of the retrieved COD (not very obvious in Figure 6 but obvious from our inversions in general) and is coherent with the asymptotic invariance of $T_b$ with increasing COD in Figure 3. This example suggests that the dynamic evolution of cloud particle properties provided by continuous temporal analysis can lend support in helping to understand cloud dynamics and more specifically in discriminating TIC1 and TIC2 particles. In the latter case the passively retrieved evidence for progressively increasing values of COD and $D_{eff}$, supported by the lidar and

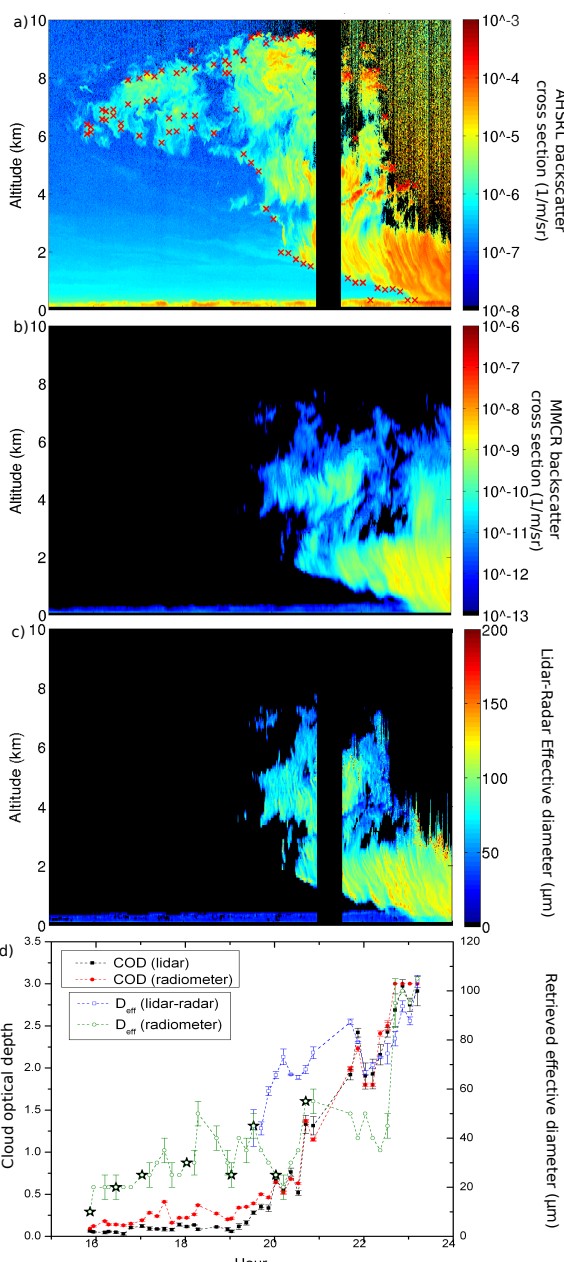

**Figure 6.** Evolution of validation and retrieval cloud parameters during a particular precipitating cloud event (January 13th, 2009) at Eureka. The error bars of the bottommost graph represent the retrieval errors. The error for the lidar COD retrievals is sufficiently small to be obscured by the size of the symbols representing this component. The red crosses, in the uppermost plot, are the cloud boundary limits used as input to our method. The points upon which a star has been superimposed satisfied the criteria defined at the beginning of section 6.2 and are thus an example of points accepted in the validation part of this article.

**Table 3.** Confusion matrix of the TIC1/TIC2 classification compared to lidar-radar retrievals. The term "err. comm." stands for the error of commission.

| | | lidar-radar | |
| --- | --- | --- | --- |
| | nb of observation: 150 | TIC1 (nb: 50) $(D_{eff} \leq 30 \, \mu m)$ | TIC2 (nb: 100) $(D_{eff} > 30 \, \mu m)$ |
| radiometer | TIC1 | 39 | 15 |
| | TIC2 | 11 | 85 |
| | Overall accuracy = 83% | err. comm. TIC1 = 22% | err. comm. TIC2 = 15% |

radar data, would lend more confidence to a classification result which indicated the presence of TIC2 type particles during the latter part of the day.

## 6.2  Validation of Our Retrieval Algorithm

Figure 7 shows COD and $D_{eff}$ comparisons between the radiometric retrievals and the combined AHSRL and MMCR re-
trievals for over 150 ice clouds observed between September 2006 and March 2009. The selection of the 150 cases was driven by different criteria: a requirement for monolayer clouds; a cloud thickness greater than 200 m (to equal or exceed the MOD-TRAN vertical layer thickness of 200 m); that the time difference between two samples be more than 30 minutes; that the clouds were non-precipitating; a subjective criterion of cloud homogeneity; a constraint whereby the evidence for cases of TIC1 only, TIC2 only or a combination of the two was determined by whether the cloud was detected by the lidar and the
radar; a requirement that the cloud be semi-transparent (AHSRL optical depth < 3); a constraint that the IR signal in any band not be saturated; that the visible optical depth should be greater than 0.1 and that the visible optical depth across the first two kilometers (where diamond dust particles are very often present in winter) should be less than 0.1. As an illustration of the influence of these criteria, only 8 of the 41 points seen in Figure 6 were selected to be part of the 150 cases.

Those clouds are not meant to be representative of the Eureka cloud climatology. Their mean base altitude (5.2 km) and vertical
extent (2.3 km) is substantially higher than a cloud climatology that was generated across four years of data (Shupe et al., 2011) (1.8 km and 2 km respectively) as well as the CALIPSO/CloudSat and ground-based climatologies on the vertical distribution of ice-only cloud (Blanchard et al., 2014).

The COD results (Figure 7a) show a significant correlation with lidar ($R^2$ = 0.95) over a large optical depth range (from 0.1
to 2.6). This level of agreement confirms the relatively strong sensitivity of the ensemble of the radiometer bands to the COD (c.f. Figure 3a). As seen in that figure, the $T_b$ sensitivity decreases with increasing COD such that the asymptotic behavior of the COD variation tends towards an upper limit of COD detectability of 2 -3 (where the spread of $T_b$ values $\approx$ the measured uncertainty in those $T_b$ values).

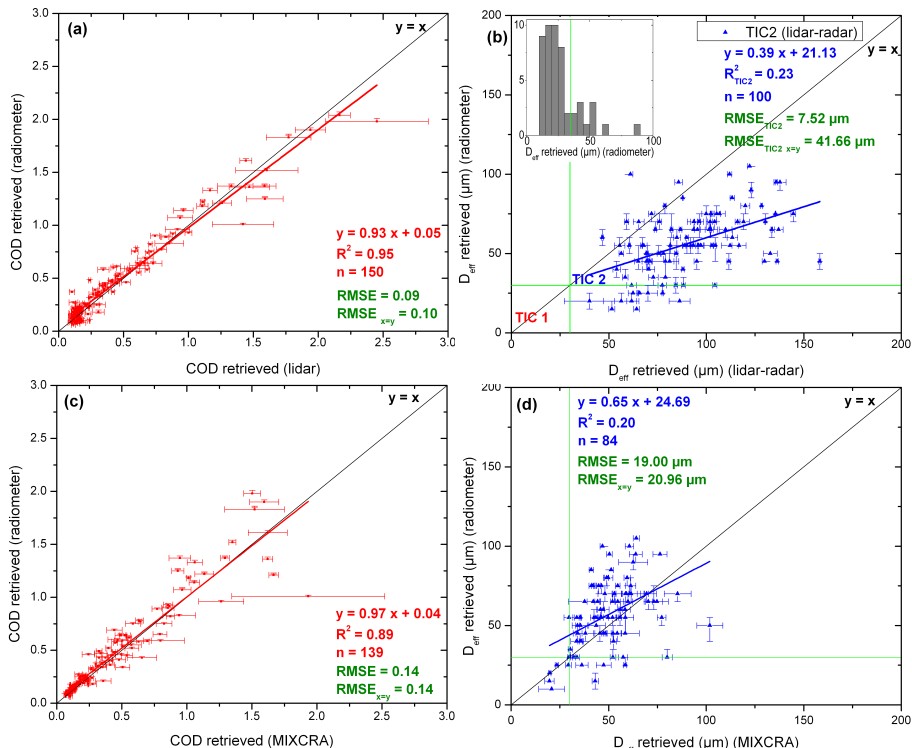

**Figure 7.** Radiometer-based retrieval results for (a) CODs compared with lidar derived CODs and (b) $D_{eff}$ compared with the lidar-radar retrieval product. The same comparisons, with MIXCRA retrievals being the reference, are shown in graphs (c) and (d). TIC1 results, because of the lidar-radar insensitivity of these small particles, are excluded from the scattergram but their $D_{eff}$ frequency distribution is shown in the inlaid histogram. The $D_{eff}$ comparisons with the MIXCRA retrievals show fewer points than the comparisons with the lidar-radar retrieval because the MIXCRA retrievals of $D_{eff}$ have relatively large uncertainties for cases where COD $\leq 0.2$ (and thus are not shown).

The quality of the particle size retrieval was difficult to quantify because the thermal IR channels become increasingly less sensitive to particles larger than $\approx 100$ μm as suggested in Figure 2b. This insensitivity to large particle sizes likely contributes to the large dispersion (and hence the marginal correlation) of retrieved TIC2 values seen in Figure 7b. The separation between TIC1 and TIC2 particles was effected based on the $D_{eff}$ values from the lidar-radar retrieval: the lower bound TIC2 value / upper bound TIC1 value was set at 30 μm. We show below that while the radiometer $D_{eff}$ retrievals are of significantly less amplitude than the lidar-radar retrievals, the 30 μm crossover criterion from TIC1 to TIC2 is sufficiently well delineated to achieve acceptable classification accuracy. One factor that complicated the $D_{eff}$ comparison over all particles sizes was the MMCR sensitivity limit at smaller particle sizes ($\approx 30$-40 μm) and the constraints this imposed on the lidar-radar retrieval. For that reason, the TIC1 results were not considered in the $R^2$ statistics. However, the TIC1 frequency distribution derived for our retrieval algorithm (inlaid histogram in Figure 7b) confirms the robustness of the retrievals inasmuch as 78% of the retrieved TIC1 population have a $D_{eff}$ less than 30 μm (and 96% less or equal to 50 μm).

**Table 4.** Confusion matrix of the TIC1/TIC2 classification compared to MIXCRA retrievals. The term "err. comm." stands for the error of commission.

| | | MIXCRA | |
|---|---|---|---|
| nb of observation: 84 | | TIC1 (nb: 7) $(D_{eff} \leq 30 \, \mu m)$ | TIC2 (nb: 77) $(D_{eff} > 30 \, \mu m)$ |
| radiometer | TIC1 | 6 | 9 |
| | TIC2 | 1 | 68 |
| Overall accuracy = 88% | | err. comm. TIC1 = 14% | err. comm. TIC2 = 12% |

The lack of TIC1 sensitivity of the lidar-radar combination means that our radiometric retrieval algorithm cannot be verified with the lidar-radar retrieval in the TIC1 particle-size region. Nonetheless, the lidar-radar classification scheme of Section 3 can at least separate out TIC1 and TIC2 cases. Table 3 presents the retrieved results in terms of the TIC1/TIC2 classification compared with the validation data. A threshold value of 30 μm was used to discriminate between the two classes in the case

of the radiometer retrieval. Those cases for which $\beta_{radar}$ was less than $10^{-15}$ $m^{-1}.sr^{-1}$ were classified as TIC1 cloud (this cutoff is illustrated by the dark regions of the $\beta_{radar}$ plot seen in the case study of Figure 6). The classification yielded satisfactory results with an overall accuracy of 83%. The TIC1 retrievals were associated with a 22% detection (omission) error $(11/50x100)$ while the TIC2 omission error was 15%. We should note that the classification results are moderately sensitive to the threshold $D_{eff}$ value assumed between the TIC1 and TIC2 classes: for threshold values of 35 and 40 μm, the overall

accuracies were respectively 82% and 82% with moderately smaller TIC1 omission errors (18% and 12% respectively).

A comparison with the MIXCRA retrievals (Figures 7c and 7d and Table 4) amounts to a coherency check between the two passive inversion techniques. While limited in terms of absolute validation, this comparison effectively reduces the array of confounding influences that can affect the retrieval quality of both approaches (and in so doing, permits a more direct evalua-

tion of the strengths and weaknesses of either technique). In Figure 7c, the good correlation between MIXCRA's and our COD retrievals and a slope near 1 confirm the robustness of the COD retrieval. The good COD correlation is expected inasmuch as a similar degree of correlation between MIXCRA and AHSRL results for ice-only clouds was previously observed (Turner and Eloranta, 2008). The comparison of the $D_{eff}$ values from our radiometer retrieval and the MIXCRA retrieval (Figure 7d) shows a somewhat better absolute agreement relative to the comparisons of our radiometer retrieval with the lidar-radar

retrieval (point scatter closer to $y = x$ but a value of $R^2$ which is also at the margins of significance).

Figure 8 indicates that all of cases with COD > 1 were classified as TIC2 for the passive and active retrievals. Clouds classified as TIC1 are, in contrast, preferentially associated with COD less than 0.3. The degree of TIC1/TIC2 classification coherence between the lidar-radar and the radiometer retrievals, in the histogram of Figure 8, illustrates the value of our semi-qualitative

(binary) classification approach for an application (the DGF effect of Blanchet and Girard, 1994) that specifically requires such

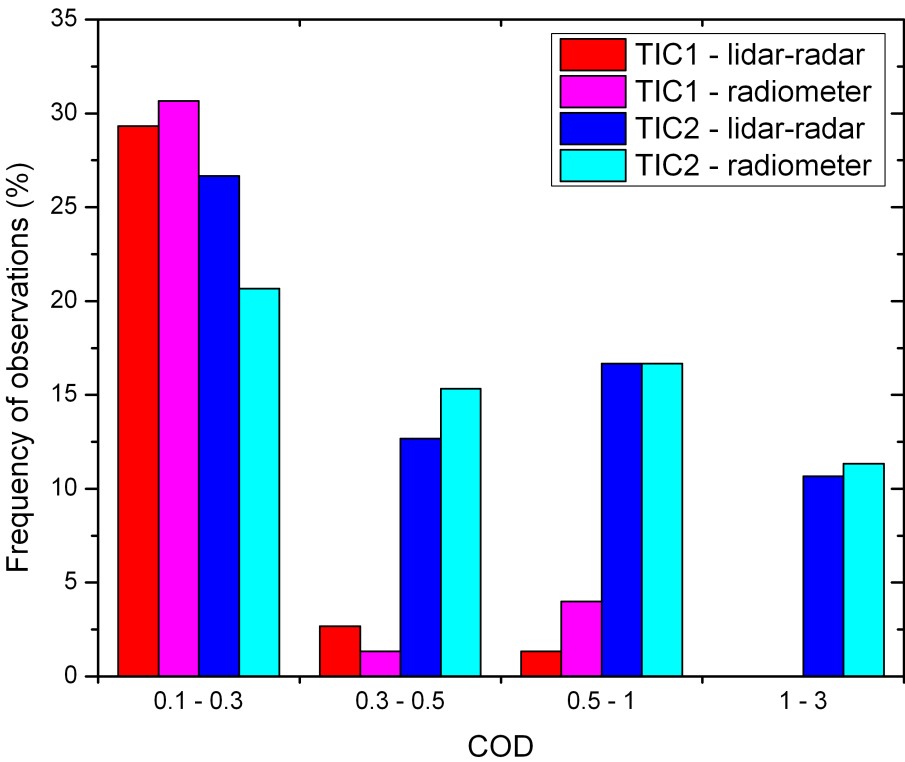

**Figure 8.** Histogram of active and passive classifications of TIC (TIC1 in red and pink; TIC2 in blue and cyan) as a function of COD.

binary information.

In order to better understand the physical implications of the retrievals, we plotted, in Figure 9, passive TIC1/TIC2 discrimination results along with downwelling longwave cloud radiative forcing (DLCRF). These data represent the 150 cloud cases

5  that we employed above for the retrieval validation. The general distribution of the DLCRF is similar to the Figure 4 results of Cox et al. (2014) who applied the MIXCRA algorithm continuously (without the separation into specific events) during the same period. As DLCRF is closely linked to the COD, one can note that the TIC1 generally have a small DLCRF of less than 10 $W.m^{-2}$. The DGF impact of the TIC2 particles (meaning their progressive removal by precipitation) would accordingly be that their radiative forcing influence would progressively decrease with an attendant cooling due to a reduction in thermal

10  interaction with the remaining TIC1 particles (and the unprecipitated TIC2 particles). A long-term analysis would help to support modeling conclusions on the impact of acid-coated ice nuclei on Arctic cloud as reported by Girard et al. (2013). These authors reported a mean downward longwave (negative) radiation anomaly at the surface of -3 -5 $W/m^2$, close to Eureka.

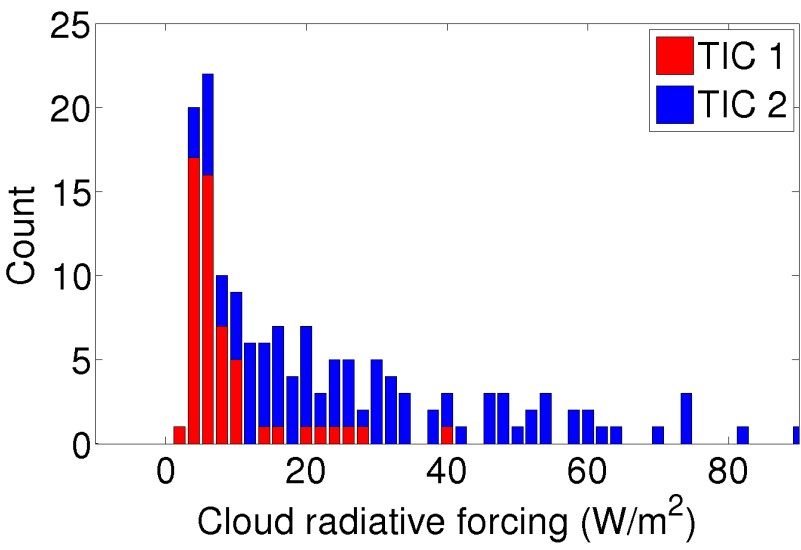

**Figure 9.** Downwelling longwave cloud radiative forcing of the 150 cases decomposed in TIC1 and TIC2

### 6.3 Additional Sensitivity Studies

The effect of particle shape on the retrievals was analyzed by re-applying the retrieval algorithm to the 150 thin ice clouds using particle shapes other than our solid column crystal standard (not shown). Retrievals results showed that the shape employed in the LUT generation has only a small influence in the performance of our retrievals (as one could have inferred from Figure 4).

This confirms the previous conclusion from Wendisch et al. (2007). The validation of COD retrievals, expressed in terms of RMSE, varied from 0.10 to 0.11 as a function of shape while overall classification accuracy varied from 82% to 83% (versus the results of 0.10 and 83% respectively for the solid column crystal shape).

We exploited the extinction coefficient profile retrieval obtainable from equation (4) in order to evaluate the impact of the

10 extinction profile as an added dimension of input to the radiative transfer model. Rather than assuming a constant extinction coefficient profile across the width of the cloud, we broke the cloud into vertical segments and obtained a coarse vertical profile. Retrieval results showed a moderate impact of the extinction profile since the RMSE is 0.09 instead of 0.10 while the overall classification accuracy remained at 83%. A similar comparison from MIXCRA results (not shown here) confirms the moderate impact of the use of real extinction profiles.

## 7 Conclusions

We developed a simple inversion technique to retrieve the optical depth and effective particulate diameter of Arctic thin ice clouds from a multi-broadband thermal radiometer using lidar and radiosonde measurements as auxiliary inputs to the inversion

routine. Specific validation elements were extracted from the combination of lidar and radar data, as well as AERI data. Our retrieval technique was applied to 150 thin ice clouds measured at the PEARL observatory (Nunavut, Canada).

The results of this study demonstrate the potential for retrieving key ice cloud parameters from thermal IR radiometry (with bands between 8 and 13 μm). The COD retrieval algorithm showed good agreement with the validation COD obtained from integrated lidar profiles. The retrievals of $D_{eff}$ showed marginal correlation for particle sizes restricted to the TIC2 category (a constraint that was driven by the insensitivity of the lidar-radar retrievals to small cloud particles). This is likely due to the weak sensitivity of thermal IR measurements for particles larger than $\approx 100$ μm. However, a classification of thin ice clouds in terms of TIC1 and TIC2 classes, using a threshold discrimination of 30 μm results in a significant classification accuracy of 83% for our passive retrieval algorithm. Further analysis showed that the extinction profile and particle shape had relatively weak impact on the retrieval results. Comparisons with the MIXCRA algorithm confirm the robustness of the optical depth retrieval.

An important application of our work would be to deploy this technique as part of a network of low-cost and robust instruments to monitor Arctic clouds. Because their occurrence, type and altitude are spatially inhomogeneous (according to Eastman and Warren, 2010; Shupe et al., 2011), we believe that additional ground-based stations would be helpful to broaden our knowledge of arctic ice clouds. Aside from being a ground-based retrieval approach in its own right (in tandem with a lidar system), this method can also be used for comparison with CALIPSO's level 2 products. The CALIPSO remote sensing suite technique employs an onboard imaging IR radiometer and the CALIOP lidar to enable the retrieval of particle size and optical depth across a narrow swath image (Garnier et al., 2012). Our retrieval, viewed as a CALIPSO validation technique is rendered all the more interesting because of the geographic position of the PEARL site; it is a high-Arctic site that sees frequent thin-cloud events and its position near the maximum latitude of the CALIOP polar orbit ensures that there are frequent overpasses of that sensor package (within a radius of hundreds of km).

**Appendix A:  Optimal estimation method**

The optimal estimation method (OEM, Rodgers, 2000) is an efficient solution to inverse problems, especially in atmospheric science. A good understanding of the technique and its associated errors is a prerequisite for the proper use of this method. Inasmuch as our application of OEM is very similar to Sourdeval's (Sourdeval et al., 2013), we used the same formalism to define the OEM components. As set out in Section 5, the OEM goal is to retrieve state variables having the maximum probability of occurrence by minimizing a cost function $\phi$:

$$\phi = (y - F(x))^T S_e^{-1} (y - F(x)) + (x - x_a)^T S_a^{-1} (x - x_a) \tag{A1}$$

where $F$ is the forward model, i.e. radiative transfer computation in our case. The state ($x$), a priori ($x_a$) and measurement ($y$) vectors are defined as:

$$x = \begin{pmatrix} D_{eff} \\ COD \end{pmatrix} ; x_a = \begin{pmatrix} D_{eff\_a} \\ COD_a \end{pmatrix} ; y = \begin{pmatrix} T_{b\_8.4} \\ T_{b\_8.7} \\ T_{b\_9.2} \\ T_{b\_10.7} \\ T_{b\_11.3} \\ T_{b\_12.7} \end{pmatrix} \tag{A2}$$

The a priori vector is the prior knowledge of the state vector, and typically corresponds to climatological values of the state vector components. In our case, the reference case of Table 2 was used to define the a priori vector and its covariance matrix. Even if any particular a priori vector values have an impact on the retrievals, it is common to attribute large uncertainties to them in the covariance matrix $S_a$ in order to let the measurement vector be the dominant driver of the retrieval. This covariance matrix is given by:

$$S_a = \begin{pmatrix} \sigma^2_{D_{eff\_a}} & 0 \\ 0 & \sigma^2_{COD_a} \end{pmatrix} \tag{A3}$$

The total error covariance matrix $S_e$ is the quadratic sum of the measurement error covariance matrix $S_y$ and the forward model parameter uncertainty covariance matrix $S_f$. We assumed that the components of the measurement or state vectors are independent (i.e., that the covariance matrix is diagonal). The measurement errors depend on the accuracy of the radiometer (which is assumed to be 0.1 K for each band, Brogniez et al., 2003). We presumed the measurement errors are wavelength-independent. The forward model errors represent the quadrature sum of the uncertainties of each input parameter (cloud base height, thickness, water vapor content and particle size) of the MODTRAN calculation. We then use the sensitivity study, Figure 4, to define the standard error ($\sigma/\sqrt{1000}$) of each parameter from the stochastic analysis. The components of $S_e$ are close to 0.30 K (between 0.28 and 0.34 K) and of the same order of magnitude as $S_y$.

*Acknowledgements.* We would like to thank the Canadian Network for the Detection of Atmospheric Change (CANDAC), Study of Environmental Arctic Change (SEARCH) and Environment and Climate Change Canada for their operational support. We are grateful to Von Walden, of the Washington State University, for the quality checked P-AERI data. This research was supported by the Natural Sciences and Engineering Research Council of Canada (NSERC), Fonds Québécois de la Recherche sur la Nature et les Technologies (FQRNT) and the Canadian Space Agency (CSA). This work was also partially supported by the U.S. Department of Energy Atmospheric System Research Program DE-SC0008830.

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
