# Peer review of "Thin ice clouds in the Arctic: Cloud optical depth and particle size retrieved from ground-based thermal infrared radiometry"

_Atmospheric Measurement Techniques, 2016_

## Referee Comment (RC1) · Anonymous Referee #1 · 26 Dec 2016

General comments: The paper titled 'Thin ice clouds in the Arctic: Cloud optical depth and particle size retrieved from ground-based thermal infrared radiometry' presents a new retrieval algorithm to estimate cloud optical depth and separate TIC1 vs TIC2 clouds based on the effective particle diameter. The paper contributes to the remote sensing field, is within the scope of AMT, and builds upon existing work that is well-referenced but some additional details are required. Results and conclusions are presented clearly and overall the paper is well-structured. Prior to publication I have several comments which need to be addressed:

Specific comments:

-P.1-2: the introduction is well-structured but fairly brief. Please consider highlighting

the relevance and importance of these observations to other communities (satellite, modelers, etc.) by describing additional applications (e.g., reference the satellite cloud climatology project Klein and Jakob, 1999; Webb et al., 2001). Also, there have been previous studies using similar or even the very same instrumentation (FIRR, AERI) to measure the radiative effect of thin ice clouds. Describing these studies demonstrate the novelty of this paper's retrieval algorithm. See for instance Libois et al., 2016 (AMT), Blanchet et al., 2011 (SPIE), Mariani et al., 2012 (AMT), and related studies therein.

-P. 2 l. 1-2 and l. 22-24: these statements require references.

-P. 3 l. 1-5 these two statements require references.

-P. 3 l. 11: Table 1 only lists the instruments used in this study. There are many more instruments operating at Eureka. Please clarify this.

-P. 4 l. 5: is there a reference for the CIMEL? An instrument paper is needed for the reader to understand the technical capabilities of the instrument.

-P. 5 l. 6: why are the 10.2-10.9 and 11.8-13.2 channels not centered at the midpoint, but the other channels are? Please also clarify whether the exact same spectral ranges were used for the integrated P-AERI spectra.

-P. 5 l. 9-10: what are the implications of using P-AERI spectra to simulate CIMEL spectra? The impact of different spectral resolution, brightness temperature accuracy, instrument noise, and sampling time should be discussed. For instance, FIRR vs. AERI brightness temperature observations have statistically significant differences, possibly due to thermal affects. Is it possible to include results (if any) that indicate the level of agreement between the CIMEL and an AERI?

-P. 7 l. 9-11: the reference case listed in Table 2 has different cloud base height and thickness values than what is listed in the 'average' row, but in the paper it is stated that the reference case was the set of mean parameters. Please clarify.

-P. 7: the reference case was for Deff = 50 microns, which is a TIC2 cloud. Please

comment on results for a TIC 1 cloud with Deff < 30 microns.

-P. 7: this analysis is heavily dependent on MODTRAN'S ability to accurately simulate these cloud properties. Please comment on MODTRAN's reliability in this regard.

-P. 8 l. 20: WVC in the Arctic has a large influence on thermal IR measurements depending on the spectral region (e.g., large influence at 20 microns) and season. Please clarify this.

-P. 9-10: the discussion of errors requires extensive elaboration, particularly in order to defend the statement on p. 9 l. 2-4. For instance, the use of radiosonde data introduces several issues which need to be addressed, including: 1) dry bias, 2) impact of using soundings during cloud cover vs. clear sky on the retrieval, 3) interpolation of the +/- 12 hour radiosonde profile. Errors associated with the OEM retrieval, such as the Sa, Se, and error covariance matrices, should be described (perhaps in the appendix) to provide a sense of the magnitude of these errors. The a priori and its covariance matrix must be carefully selected due to their large impact on the retrieval's outcome – more detail is needed here.

-P. 11 l. 9: please describe why this tolerance value was used.

-P. 11 l. 29: there are several papers that state the wavelength range of the P-AERI is up to 20 microns. Please clarify this discrepancy.

-P. 14 l. 14: on p. 10 it is states that upper and lower cloud boundaries are obtained where 7 vertical samples (52.5 m) comply with the backscatter requirement. Please clarify this discrepancy (52.5 vs. 200 m).

-P. 15 l. 1-2: please clarify what is meant by "sufficiently accurate."

-It would be interesting to see a comparison between MIXCRA and the Lidar-radar retrieval to provide a sense of how well the two established methodologies compare.

-P. 16 l. 15-on: please clarify whether "the comparison" is between MIXCRA and the

AHSRL Deff observations in the Turner and Eloranta paper and state their R-squared result.

-The results discussed regarding Fig. 9 are important and yet left out of the conclusion – consider including them.

Technical corrections:

-P.2 l. 8-9: the word 'properties' is used three times in ∼12 words.

-P.2 l.18-19: please consider reordering your examples so that the references are listed in chronological order.

-P.2 l. 27-29: the referenced work is not 'recent,' as your sources are 22 and 7 years old.

-P. 3 Eqn. 2: you have explained all variables except $D^2$.

-P. 4 Fig. 1: no date and observation time is provided in the figure caption. There is also no color bar legend and units.

-P. 5 l. 12: the list of references is not complete. If you are citing examples of studies, then state this using 'e.g.,'

-Fig. 2 and Fig. 4: the x-axis label 'lambda' should be changed to reflect what the physical quantity is, i.e., 'Wavelength.' The figure caption should clearly indicate whether this is simulated or observed values.

-Several places in the paper are missing a space. For instance, Table 2 caption, p. 7 l. 5, caption of Fig. 4.

-The text at the bottom right of Fig. 3 (b) should be moved into the figure caption.

-Fig. 4: please state the sample size of the simulation both in the figure caption and your discussion.

-P. 11 l. 10: "mean't" should be "meant."

-P. 12 l. 7: avoid nested brackets if possible.

-P. 12 l. 12: "and demonstrate that our retrieval produces give physically" – please fix.

-Fig. 6: titles should be moved next to the color bar (right-hand side). It should be clearly indicated that Fig. 6 (c) is Lidar-Radar retrieval in the figure and/or in the caption.

-Fig. 6: it is stated that only 8 of the 41 points from (b) passed the screening test. Please identify these points in the figure (perhaps a different color) and explain the distinction in the figure caption.

-Fig. 7: the equation (y=...) is missing in (b). The RMSE calculations are not shown in (b) and (d). The caption is also quite long – try to shorten and move parts to the discussion.

-P. 20 l. 16: 'Environment Canada' is now 'Environment and Climate Change Canada.'

-P. 22 l. 11-15 and p. 24 l. 10-13: these references are out of order based on publication year.

---

## Referee Comment (RC2) · Anonymous Referee #2 · 12 Jan 2017

This paper describes a novel method to determine (thin) ice cloud optical depth as well as some limited mean particle size information using a combination of lidar data, IR radiometry and atmospheric thermodynamic state information. The technique appears robust and appears reasonably easy to implement. It could also be a candidate for network deployment.

The paper is, in the main, clear and well-written and worthy of publication. Another reviewer has already noted a number of issues. I have noted a further few aspects that should be improved before final acceptance.

[Figure]

P1: line 1: What type of profile information ? Please be specific here.

P11: Lines 25-28: The authors should expand the lidar multiple-scattering discussion. It is not quite satisfying/convincing to me. I agree that the application of Eloranta's formalism is appropriate, (it would be useful if they specified the equation number of the formula they used) however, they must follow through to discuss the errors in terms of optical depth and not leave the discussion solely in terms of Pt/P1. Also, the discussion must be made much clearer.

For example, (looking at Eq. 4) it is clear that the important factor in determining the COD is the ratio of the signals at cloud base and cloud top. Thus, the relevant quantity for determining the effect of MS is the Pt/P1 ratio at cloud top (since at cloud-base Pt=0). However, curiously, the worst-case cloud-base height is defined as 5.5 km but the cloud top altitude is not specified.

Assuming that the worst-case Pt/P1 ratio value of 60% quoted by the authors is indeed the value at cloud-top. The error in COD induced by MS effects for the worst-case scenario can be easily calculated. Using Eq.4 it is easy to show that the effect of MS will be to lower the retrieved extinction by an amount given by

dCOD_ms = -0.5 log (1.0+Pt/P1)

and Pt=0.6 P1 implies that dCOD_ms= 0.23 which is about a -10% bias.

Moving on, it is not clear what the authors mean by the "overall average value for the Pt/P1 upper limit". Is that the altitude averaged value for the just described "worst-case" scenario or the average ratio at cloud-top over the investiged cases ? If the former, then it is not a useful quantity. If the latter then that would correspond to a COD retrieval bias of -0.05 which could be significant for many of the results presented in this paper (e.g. see Fig .6 before about 19:00). Indeed accounting for the MS effects may bring the IR radiometer and lidar results more into line with each other values.

The authors have enough information to define the COD, cloud geometry and particle

size ranges they are dealing with. Using this information and Eloranta's model I think with not too much effort they could define and apply a mean eta factor as was done in the paper by Platt which they reference.

Page 20: Line 5-10: Can the authors please comment on how realistic it is to assume that all the measurement errors on the IR BTs are indeed independent ?

Page 20: Line 14: sigma/sqrt(1000). What is the significance of the sqrt term ?

---

## Referee Comment (RC3) · Anonymous Referee #3 · 13 Jan 2017

Review of "Thin ice clouds in the Arctic: Cloud optical depth and particle size retrieved from ground-based thermal infrared radiometry" by Blanchard et al.

General comments

This paper reports on the development and testing of new LIRAD method that utilizes thermal infrared bands available on the CIMEL CE-312 radiometer, which is considerably cheaper than the AERI. The latter has been used in the recent past to provide the thermal components to retrieve COD and Deff from the surface. The implicit motivation for this study seems to be that the LIRAD method can be used more often if the CIMEL-312 could be used instead of the AERI, since it is attempting to do the same things that AERI already provides. The explicit motivation is that the ability to discriminate between small and large Deff values would help the study of aerosol/cloud interactions in polar regions, especially regarding aerosol influence on precipitative cooling and (as inferred from the DLCRF computations) on radiative heating of the surface. The writing is of average quality, but has some grammatical issues. The paper is scientifically sound in its approach, but the results and method do not appear to be particularly new.

The algorithm is new in that it uses specific bands not previously employed in the AERI-LIRAD approach. However, they are not all that different from the AERI microwindows. Except for lacking the wavelengths longer than 14 µm, this is essentially the same algorithm.  If not, then it needs more contrasting with the MIXCRA. It relies on the phase being known already, whereas, I believe, the longer wavelengths used in the MIXCRA were primarily for phase discrimination. It is not surprising then that the results are quite close to those from the MIXCRA. The bottom line that I believe the authors should address is "how many wavelengths are actually needed to replace the 19 microwindows of MIXCRA having lambda < 13 µm to perform the retrieval?" Could you do it with 2 or 3 channels? There appears to be a lot of information redundancy in the bands that were used.

The validation effort includes the comparisons with MIXCRA (noted above) and with lidar for COD. This begs the question: If the lidar is *required* for the LIRAD method and it produces a reliable COD (it is used as a reference), why then is the IR method used to estimate COD?  Why not simply estimate Deff using the IR data and the lidar COD as input? Lidar retrievals of COD are quite common for most lidars deployed for surface observations. What am I missing? How will phase be determined if this approach is implemented elsewhere?

I think this paper can be published, but it needs some major revisions to provide better justification as to why it is necessary and to flesh out the analysis by addressing the questions above.

Minor comments

P1, L24: "temperatures" does not belong here

P3, L4: " aerosols acting as ice and water cloud nuclei, cloud microphysics, precipitation and radiation" does not make any sense. Please reorder this so that aerosols do not appear to act as cloud microphysics.

P3, L14: You may want to modify the sentence with "could possibly lead to" or something similar, since the "dehydration greenhouse feedback" is only a proposed mechanism.

P4, L3: Please "COD" for singular and "CODs" or "COD values" for plural here and throughout the paper

P4, L9: "northern most" should be one word

P4, L14: if the same summary is given by both references, then leave one out. If two different summaries are provided, then change everything to the plural form.

P4, L17: "in the order of" should be "on the order of"

P5, L24-25: The <2% refers *only* to downwelling radiation viewed at the zenith. It can be up to10% for other viewing conditions, particularly for upwelling radiation (e.g., Minnis et al., JAS 93). The viewing limitation should be highlighted again when referring to the 2%.

P7,L16: COD can only have an amplitude if you are referring to its oscillation with time or space. Otherwise, please refer to it as magnitude or value.

P8, L3-6: Sentence should be broken up for clarity.

P8, L16: cloud altitude and thickness uncertainties do not have amplitudes in this context. See comment above.

P10, L1: Here and elsewhere, the form of the modifier does not need to match that of the noun. Should be "ice cloud retrievals".

P11, L10: "mean't" appears to be a typo.

P12, L12: "produces gives", please use one or the other

P12, L18: should be "hydrometeor diameter"

P16, L4-5: "ommission" has only one "m"

---

## Author Comment (AC1) · 12 Mar 2017

Response to Anonymous Referee #1

The reviewer's comments are in black and our answers are in blue. Snippets of text from the submitted manuscript are in italics while modifications of the manuscript are shown in bold italics. The pages and lines reported here correspond to the submitted manuscript.

General comments: The paper titled 'Thin ice clouds in the Arctic: Cloud optical depth and particle size retrieved from ground-based thermal infrared radiometry' presents a new retrieval algorithm to estimate cloud optical depth and separate TIC1 vs TIC2 clouds based on the effective particle diameter. The paper contributes to the remote sensing field, is within the scope of AMT, and builds upon existing work that is well-referenced but some additional details are required. Results and conclusions are presented clearly and overall the paper is well-structured. Prior to publication I have several comments which need to be addressed:

We are grateful to this reviewer for the helpful comments. We provide below a point-by-point reply to his comments.

Specific comments:
-P.1-2: the introduction is well-structured but fairly brief. Please consider highlighting the relevance and importance of these observations to other communities (satellite, modelers, etc.) by describing additional applications (e.g., reference the satellite cloud climatology project Klein and Jakob, 1999; Webb et al., 2001).
Also, there have been previous studies using similar or even the very same instrumentation (FIRR, AERI) to measure the radiative effect of thin ice clouds. Describing these studies demonstrate the novelty of this paper's retrieval algorithm. See for instance Libois et al., 2016 (AMT), Blanchet et al., 2011 (SPIE), Mariani et al., 2012 (AMT), and related studies therein.

We have added more explanations about the relevance of ice clouds study in the introduction.
P1 L17: *Predictions of future climate change and its regional and global impacts require that a better understanding of the radiative transfer interactions between clouds, water vapor and precipitation be incorporated into appropriate models. **Recent model intercomparisons indicate large variability in ice cloud parameters (for example ice water content) amongst high-latitude models in the framework of CMIP5 (Coupled Model Intercomparison Project) (Jiang et al., 2012). Shortcomings in ice cloud parametrization (Baran, 2012) impact their representation of radiative effects as well as water cycles and leads to uncertainties in quantifying cloud feedbacks in the context of climate change (Waliser et al., 2009).** High-altitude thin ice clouds consisting of pure ice crystals, which cover between 20 to 40% of the Earth (Wylie and Menzel, 1999), can, for example, have opposing effects on the radiative properties of the Earth.*

*P2L6: **The advent of active sensors onboard satellites (for example CALIPSO/CloudSat) has enabled the application of considerably more resources for polar region ice cloud studies. This permits the evaluation of climate models (Jiang et al, 2012) and satellite cloud climatologies (Sassen et al., 2008). Long-term ground-based observations which are also essential for the validation of models and satellite climatology are, however, limited in their Arctic coverage (Heymsfield, 2017).***

*P 2 L 21: **An instrument designed to study thin ice clouds in the Arctic from space using far and thermal infrared channels (Blanchet et al., 2011) was recently tested during an airborne campaign in the High Arctic (Libois et al, 2016).***

-P. 2 l. 1-2 and l. 22-24: these statements require references.

P. 2 l. 1-2: *The macrophysical and microphysical properties of thin ice clouds determine which process dominates and hence determine the net forcing of thin ice clouds on the climate system **(Stephens, 2005)**.*
P. 2 l. 22-24: *In this paper, we examine how multi-band thermal measurements of zenith sky radiance can be used to retrieve what are arguably the most critical extensive and intensive parameters influencing the radiative effects of ice clouds **(as in the early work on clouds from Nakajima and King, 1990)**: cloud optical depth (COD) and effective particle diameter ($D_{eff}$).*

-P. 3 l. 1-5 these two statements require references.

P3 L1: *Water vapor and clouds are a significant climate modeling challenge since they represent major radiative forcing influences, while being the least understood components of the climate system **(Waliser et al., 2009; Jiang et al., 2012)**.*
P3. L3: *Much of the recent research has been focused on aerosol-cloud interaction processes involving aerosols acting as ice and water cloud nuclei **and their subsequent affect on** cloud microphysics, precipitation and radiation **(see for example, Feingold and McComiskey (2016) on recent ARM campaigns, Winker et al. (2010) and Illingworth et al. (2015) respectively, on the cloud remote sensing mandate of the A-Train and EarthCARE satellite missions and Jouan et al. (2013) as part of the NETCARE project)**.*

-P. 3 l. 11: Table 1 only lists the instruments used in this study. There are many more instruments operating at Eureka. Please clarify this.

This is true. We have changed the Table 1 caption. ***List of the instruments used for this article, which is a partial list of the instrumentation inventory of the PEARL observatory.***

-P. 5 l. 5: is there a reference for the CIMEL? An instrument paper is needed for the reader to understand the technical capabilities of the instrument.

A very similar instrument (we used a newer version with 2 additional channels) was fully detailed in Legrand et al. (2000), in terms of performances and error analysis. The companion paper (Brogniez et al., 2003) focused on its behavior in field campaigns and showed radiometric measurement accuracy of about 0.1 K. We have added those references in the manuscript. P5L5 :***(see Legrand et al. (2000) and Brogniez et al. (2003) for the description of a similar instrument).***

-P. 5 l. 6: why are the 10.2-10.9 and 11.8-13.2 channels not centered at the midpoint, but the other channels are? Please also clarify whether the exact same spectral ranges were used for the integrated P-AERI spectra.

By centered, we meant the peak wavelength of the spectral filter response. This was clarified in the manuscript. P.5 L.6: *... and **have a filter response peak value**  at 8.4, 8.7, 9.2, 10.7, 11.3 and 12.7 µm.* As shown in the figure below, the peaks are not always located in the middle of the channel.

The integration over the P-AERI spectra was weighted using the spectral filter response given by the manufacturer (see figure below). This was clarified in the text:

*P.5 L.6: In actual fact we had to simulate the response of this radiometer by **convolving the spectral transmittance of each filter with the** spectra of the Eureka Polar AERI (P-AERI) instrument (provided by Von Walden at the U. of Idaho and NOAA).*

[Figure]

Figure 1: Transmittance of the 6 channels, normalized by their maximum value

-P. 5 l. 9-10: what are the implications of using P-AERI spectra to simulate CIMEL spectra? The impact of different spectral resolution, brightness temperature accuracy, instrument noise, and sampling time should be discussed. For instance, FIRR vs. AERI brightness temperature observations have statistically significant differences, possibly due to thermal affects. Is it possible to include results (if any) that indicate the level of agreement between the CIMEL and an AERI?

We agree that a comparison side-by-side of both instruments (P-AERI and CE-312) would be beneficial to better assess the performances of the radiometer. Unfortunately during field campaigns, the CE-312 wasn't ready for deployment and a slightly different instrument (CE-332), with only 3 bands, was used. The comparison with the 8.7 µm band (Figure 2 below) showed relatively good correlative agreement ($R^2$=0.97) except in the presence of low brightness temperatures (clear sky). The correlative statistics were similar for the 2 other bands (10.8 µm and 12.6 µm), resp. 0.98 and 0.99.

[Figure]

Figure 2: Brightness temperature comparison between integrated P-AERI spectra and the CE-332 8.7 µm channel during the field campaign in September 2007.

The P-AERI, which includes a highly accurate radiometric calibration system that employs 2 blackbody references, is expected to have a higher absolute accuracy (as an indicator of absolute differences relative to the P-AERI, we would note that the rms error relative to the "y = x" line on Figure 2 is 9.04 K for all the points and 5.23 K if the "Clear sky" points are not included). The noise equivalent temperature difference (NETD) is less than 30 mK, which can be compared with the value of 50 mK (at 20 °C) for the CE-312 instrument. In spite of the more moderate performance of the CE-332, we would remind the reviewer that the retrieval method incorporates the CE-312 measurement error. It is for this latter reason and the fact that we did not have a full, prototype, 6-band instrument in the field that we decided to not include a comparison of the CE-332 with the P-AERI directly in the text of the article.

-P. 7 l. 9-11: the reference case listed in Table 2 has different cloud base height and thickness values than what is listed in the 'average' row, but in the paper it is stated that the reference case was the set of mean parameters. Please clarify.

For the reference case, the cloud base height and thickness values were rounded to the nearest step of the MODTRAN vertical profiles for convenience. We added the following sentence to the legend to explain this: ***For the reference case, the cloud base height and thickness values were, for the sake of convenience, rounded to the nearest step of the MODTRAN vertical profiles***.

-P. 7: the reference case was for $D_{eff}$ = 50 microns, which is a TIC2 cloud. Please comment on results for a TIC 1 cloud with $D_{eff}$< 30 microns.

We used a reference value that is common for ice clouds (Sourdeval et al., 2013). If we take a reference $D_{eff}$ value of 15 µm, the sensitivity analysis is relatively similar with moderate differences of <~ 1 K (compare Figure 3 below with Figure 4 in the paper). We added the following sentence to the discussion of Figure 4 (P. 8 L9); ***We note that there was little sensitivity to the choice of a 50 µm effective diameter for the reference case: changing this typical TIC2 value to a value more representative of TIC1 particles produced differences in Figure 4 less than 1 K.***

[Figure]

Figure 3: Sensitivity of $T_b$ as a function of the six key radiative transfer parameters, when the reference $D_{eff}$ is set as 15 µm.

-P. 7: this analysis is heavily dependent on MODTRAN'S ability to accurately simulate these cloud properties. Please comment on MODTRAN's reliability in this regard.

We used MODTRAN4 in this article since the 1 cm$^{-1}$ resolution was adequate for our needs. Its multiple scattering capabilities are as accurate as the user requires: we employed this capability, with sensitivity tests, to ensure accurate calculations in the case of the thicker TICs. MODTRAN4 is flexible in the way it allows the user to configure the thermodynamical state of the atmosphere and the optical properties of complex scattering / absorbing constituents such as clouds. Thus, for example, it allowed us to incorporate the ice cloud properties recently parameterized by Baum et al., (2014) and to replace the standard MODTRAN vertical profiles by Eureka-specific radiosonde profiles of temperature and humidity. The layering capabilities allowed us to include cloud bottom and top height from our lidar profiles and to test the sensitivity of the radiative transfer computations to layer resolution.

-P. 8 l. 20: WVC in the Arctic has a large influence on thermal IR measurements depending on the spectral region (e.g., large influence at 20 microns) and season. Please clarify this.

We reinforced the Figure 4 evidence for weak WVC influence with the following modification of the text describing Figure 4. *P.8 L.20: Water vapor content (WVC) in the atmosphere, which remains relatively low **during the polar winter at Eureka, has a weak absorption influence on the CE-312 band**  radiance measurements .*

-P. 9-10: the discussion of errors requires extensive elaboration, particularly in order to defend the statement on p. 9 l. 2-4. For instance, the use of radiosonde data introduces several issues which need to be addressed, including: 1) dry bias, 2) impact of using soundings during cloud cover vs. clear sky on the retrieval, 3) interpolation of the +/- 12 hour radiosonde profile. Errors associated with the OEM retrieval, such as the Sa, Se, and error covariance matrices, should be described (perhaps in the appendix) to provide a sense of the magnitude of these errors. The a priori and its covariance matrix must be carefully selected due to their large impact on the retrieval's outcome – more detail is needed here.

1) The Vaisala radiosondes are known to be subject to dry bias which tends to underestimate the relative humidity by 2-8% (Wang et al., 2013), especially in dry conditions, which could be problematic in an arctic environment. This bias is less severe during nighttime (Turner et al., 2003) and by extension during the Arctic Winter. Some authors (Treffeisen et al., 2007; Rowe et al., 2008) have studied the bias in polar regions and have shown the bias could be up to − 10 % in the worst conditions. However, the 6 channels of the radiometer are far less sensitive to the WVC than to COD, as one can see in Figure 4, and are in an atmospheric window of water vapor. We agree that this could be more problematic for bands in the far infrared.

P10 L6: *Radiosonde humidity sensors are known to be subject to dry bias especially in dry conditions and could underestimate the relative humidity up to 10 % (Rowe et al., 2008). The 6 channels are however far less sensitive to the WVC than to COD (see Figure 4) and therefore the bias is expected to be lower in cloudy conditions.*

2) and 3) To avoid the issue of interpolating radiosondes over extensively long periods of time, the cases were selected as close as possible to radiosonde launch times. Indeed, more than 40 % of the 150 cases occurred at a maximum of 3 hours before or after radiosonde profiles. In the case of temperature, the absolute value of the temporal variations between 2 radiosondes, averaged between 2 and 8 km, during the 3 polar winters, is about 1.72 °C, which means 0.14 °C/hour. The following sentences were added in the text: P10 L6: *To avoid the issue of interpolating radiosondes over extensively long periods of time, the cases were selected as close as possible to radiosonde launch times.*

We agree that a careful definition of covariance matrix and errors is needed in the optimal estimation method (OEM). We have added more details about the covariance matrix in the appendix A.

P 20 L13: *In our case, the reference case of Table 2 was used to define the a priori vector and its covariance matrix. … The measurement errors depend on the accuracy of the radiometer, which is assumed to be 0.1 K for each band (Brogniez et al., 2003), and we presumed the measurement errors are wavelength-independent. … The standard deviations of the components of $S_e$ are close to 0.30 K (between 0.28 and 0.34 K), on the same order of magnitude than $S_y$.*

-P. 11 l. 9: please describe why this tolerance value was used.

The value of $10^{-15}$ 1/m/sr represents approximatively the minimum detectable reflectivity close to the surface, in the MMCR general mode. The minimum detectable reflectivity is an estimate of a minimally significant value that we determined from an analysis of MMCR profiles. The following change was made to the text;... *less than $10^{-15}$ $m^{-1}$ $sr^{-1}$(an empirically determined value of minimum detectability) were eliminated from any ...*

-P. 11 l. 29: there are several papers that state the wavelength range of the P-AERI is up to 20 microns. Please clarify this discrepancy.

It is true that the original P-AERI was designed to acquire measurements up to 20 µm (the measurements become noisy after 20µm). The text in the manuscript was changed to (P11 L29): *Although intervals of P-AERI spectra (wavelength range of 3 - 20 µm)...*
More recent versions of the AERI (for example the Extended-AERI, see Mariani et al., 2012), have more far infrared capabilities.

-P. 14 l. 14: on p. 10 it is states that upper and lower cloud boundaries are obtained where 7 vertical samples (52.5 m) comply with the backscatter requirement. Please clarify this discrepancy (52.5 vs. 200 m).

The vertical resolution of 7.5 m was used in a previous version of this study and is now set as 30 m to smooth the lidar and radar profiles. The value of 200 m is set as the vertical step in the MODTRAN simulations. The value of 120 m (4 x 30 m) is needed to delimit cloud boundaries and therefore used to infer the COD and $D_{eff}$ values from active instruments in the validation part of the manuscript. We didn't set the criteria to be 7 continuous pixels (= 210 m) because in some thin case (see for example in figure 6a, before 18:00), it happens that few pixels, in the vertical profile, don't match the threshold on the lidar signal.

P14 L 14: *... a cloud thickness greater than 200 m **(to comply with MODTRAN vertical step),** ...*

-P. 15 l. 1-2: please clarify what is meant by "sufficiently accurate."

By "sufficiently accurate." we meant that there are very few cases (only 2) retrieved by lidar+radar which have a $D_{eff}$ less than 50 μm. This means that the threshold between TIC1 and TIC2 (30 μm) was chosen in a conservative manner to reduce the crossover between TIC1 and TIC2. We made the following change to the text (P15 L1-2); *the **30 μm** crossover **criterion** from TIC1 to TIC2 is **sufficiently well delineated** to achieve acceptable classification accuracy*

-It would be interesting to see a comparison between MIXCRA and the Lidar-radar retrieval to provide a sense of how well the two established methodologies compare.

A comparison between MIXCRA and AHSRL/MMCR can be found below. But as this article focuses on the performances of the radiometer, we chose not to include a MIXRA and AHSRL/MMCR comparison.

[Figure]

Figure 4: MIXCRA retrieval results compared with lidar derived COD (left) and with the lidar-radar $D_{eff}$ retrieval product (right)

-P. 16 l. 15-on: please clarify whether "the comparison" is between MIXCRA and the AHSRL $D_{eff}$ observations in the Turner and Eloranta paper and state their R-squared result.

P. 16 l. 15: The term "the comparison" was referring to the comparison between our retrievals and MIXCRA (Figure 7d). In actual fact, the paper of Turner and Eloranta (2007) doesn't include comparison of $D_{eff}$ retrievals. To eliminate this source of confusion, we changed the sentence in question to; "*The comparison of the $D_{eff}$ values **from our radiometer retrieval and the MIXCRA retrieval (Figure 7d)** shows a somewhat better absolute agreement relative to the comparisons of our*

*radiometer retrieval* with the lidar-radar retrieval…"

-The results discussed regarding Fig. 9 are important and yet left out of the conclusion– consider including them.

We added these sentences at the end of section 6.2 to expand the discussion, rather than in the conclusion, because we thought it would be more relevant: P18: L2. *A long-term analysis would help to support the modeling conclusions on the impact of acid-coated ice nuclei on Arctic cloud as reported by Girard et al. (2013). These authors reported a mean downward longwave (negative) radiation anomaly at the surface of -3 -5 W/m2, close to Eureka.*

Technical corrections:

-P.2 l. 8-9: the word 'properties' is used three times in 12 words.

The sentence was rewritten: *Numerous researchers have exploited the thermal IR behavior of the absorption and scattering efficiencies  of cloud particles as a means of retrieving  COD and particle effective sizes (e.g., Inoue, 1985).*

-P.2 l.18-19: please consider reordering your examples so that the references are listed in chronological order.

This was done in the manuscript.

-P.2 l. 27-29: the referenced work is not 'recent,' as your sources are 22 and 7 years old.

The first reference is important to understand the concept of the dehydration feedback. Since, observations from space (Grenier et al., 2009), airborne campaigns (Jouan et al., 2013) as well as laboratory simulations (Chernoff and Bertram, 2010)  and climate model (Girard and Sokhandan, 2014) tends to confirm the impact of acid coating aerosols on cloud  microstructure  and  radiative  forcing over the Arctic during the cold season. In any case, we modified the sentence a bit to get rid of "recent"; "*This approach **was** motivated by **previously published** research that  indicated such a discrimination would play a key role in characterizing an important aerosol/cloud interaction process in Polar winter, namely precipitative cooling (**see, for example,** Blanchet and Girard, 1994; Grenier et al., 2009).*"

-P. 3 Eqn. 2: you have explained all variables except $D^2$.

P4 L1: *where $Q_{ext}$ is the extinction efficiency (extinction cross section per unit projected-particle-area) (Hansen and Travis, 1974)**, D is the particle diameter** and a(D) is the ice particle number density per unit increment in diameter.*

-P. 4 Fig. 1: no date and observation time is provided in the figure caption. There is also no color bar legend and units.

Corrected.

-P. 5 l. 12: the list of references is not complete. If you are citing examples of studies, then state this using 'e.g.,'

Corrected.

-Fig. 2 and Fig. 4: the x-axis label 'lambda' should be changed to reflect what the physical quantity is, i.e., 'Wavelength.' The figure caption should clearly indicate whether this is simulated or observed values.

The term "lambda" was replaced by "wavelength" in the figures.

-Several places in the paper are missing a space. For instance, Table 2 caption, p.7 l.5, caption of Fig. 4.

Corrected.

-The text at the bottom right of Fig. 3 (b) should be moved into the figure caption.

Done.

-Fig. 4: please state the sample size of the simulation both in the figure caption and your discussion.

Done. Figure 4 caption: *Sensitivity of Tb as a function of six key radiative transfer parameters. The standard deviations (in units of K) are obtained by stochastically varying, **with a sample size of 1000**, the parameters of interest within the limits given in the Table 2. P8L4: … each parameter individually varied using a random number generator, **for 1000 samples**, with a normal probability distribution …*

-P. 11 l. 10: "mean't" should be "meant."

Corrected.

-P. 12 l. 7: avoid nested brackets if possible.

Corrected. *Within the scope of this current study, the cloud-phase determination from MIXCRA is used to ensure the comparison of ice-only cloud properties (i.e. those MIXCRA retrievals that yielded negligible liquid water path,  were taken as being pure ice-cloud cases). MIXCRA results are used here as an alternative point of reference for our retrievals.*

-P. 12 l. 12: "and demonstrate that our retrieval produces give physically" – please fix.

We removed the term "produces".

-Fig. 6: titles should be moved next to the color bar (right-hand side). It should be clearly indicated that Fig. 6 (c) is Lidar-Radar retrieval in the figure and/or in the caption.

Done.

-Fig. 6: it is stated that only 8 of the 41 points from (b) passed the screening test. Please identify these points in the figure (perhaps a different color) and explain the distinction in the figure caption.

Done. The figure caption has been modified: ***The points surrounded by a star satisfied the criteria defined at the beginning of section 6.2 and were used in the validation part of this article.***

-Fig. 7: the equation (y=. . .) is missing in (b). The RMSE calculations are not shown in (b) and (d). The caption is also quite long – try to shorten and move parts to the discussion.

The equations and RMSE were added in (b) and (d). The caption was shortened.

-P. 20 l. 16: 'Environment Canada' is now 'Environment and Climate Change Canada.'

Corrected.

-P. 22 l. 11-15 and p. 24 l. 10-13: these references are out of order based on publication year.

Done.

References to be added to the article:

Baran AJ. From the single-scattering properties of ice crystals to climate prediction: a way forward. Atmos Res 2012;112:45–69.

Baum, B. A., P. Yang, A. J. Heymsfield, A. Bansemer, A. Merrelli, C. Schmitt, and C. Wang, 2014: Ice cloud single-scattering property models with the full phase matrix at wavelengths from 0.2 to 100 μm. *J. Quant. Spectrosc. Radiant. Transfer, vol. 146, 123-139.*

Blanchet, J.-P., Royer, A., Châteauneuf, F., Bouzid, Y., Blanchard, Y., Hamel, J.-F., de Lafontaine, J., Gauthier, P., O'Neill, N. T., Pancrati, O., and Garand, L.: TICFIRE: a far infrared payload to monitor the evolution of thin ice clouds, doi:10.1117/12.898577, http://dx.doi.org/20 10.1117/12.898577, 2011.

Brogniez, G., C. Pietras, M. Legrand, P. Dubuisson, and M. Haeffelin, 2003: A High-Accuracy Multiwavelength Radiometer for In Situ Measurements in the Thermal Infrared. Part II: Behavior in Field Experiments. J. Atmos. Oceanic Technol., 20, 1023–1033, doi: 10.1175/1520-0426(2003)20<1023:AHMRFI>2.0.CO;2.

Chernoff, D. I., and A. K. Bertram (2010), Effects of sulfate coatings on the ice nucleation properties of a biological ice nucleus and several types of minerals, J. Geophys. Res., 115, D20205,

Feingold, G. and A. McComiskey, 2016: ARM's Aerosol–Cloud–Precipitation Research (Aerosol Indirect Effects). Meteorological Monographs, 57, 22.1–22.15, doi: 10.1175/AMSMONOGRAPHS-D-15-0022.1.

Girard, E., G. Dueymes, P. Du and A.K. Bertram, 2013: Assessment of the Effects of Acid-Coated Ice Nuclei on the Arctic Cloud Microstructure, Atmospheric Dehydration, Radiation and Temperature during Winter. International Journal of Climatology. 33, 599-614. DOI: 10.1002/joc.3454

Girard, E. and SokhandanAsl, N. Relative importance of acid coating on ice nuclei in the deposition and contact modes for wintertime Arctic clouds and radiation, MeteorolAtmos Phys (2014) 123: 81. doi:10.1007/s00703-013-0298-9

Heymsfield, A., M. Krämer, A. Luebke, P. Brown, D. Cziczo, C. Franklin, P. Lawson, U. Lohmann, G. McFarquhar, Z. Ulanowski, and K. Van Tricht, 2017: Cirrus Clouds. Meteorological Monographs, 58, 2.1–2.26, doi: 10.1175/AMSMONOGRAPHS-D-16-0010.1.

Illingworth, A., H. Barker, A. Beljaars, M. Ceccaldi, H. Chepfer, N. Clerbaux, J. Cole, J. Delanoë, C. Domenech, D. Donovan, S. Fukuda, M. Hirakata, R. Hogan, A. Huenerbein, P. Kollias, T. Kubota, T. Nakajima, T. Nakajima, T. Nishizawa, Y. Ohno, H. Okamoto, R. Oki, K. Sato, M. Satoh, M. Shephard, A. Velázquez-Blázquez, U. Wandinger, T. Wehr, and G. van Zadelhoff, 2015: The EarthCARE Satellite: The Next Step Forward in Global Measurements of Clouds, Aerosols, Precipitation, and Radiation. Bull. Amer. Meteor. Soc., 96, 1311–1332, doi: 10.1175/BAMS-D-12-00227.1.

Jiang, J. H., et al. (2012), Evaluation of cloud and water vapor simulations in CMIP5 climate models using NASA "A-Train" satellite observations, J. Geophys. Res., 117, D14105, doi:10.1029/2011JD017237.

Jouan, C., Pelon, J., Girard, E., Ancellet, G., Blanchet, J. P., and Delanoë, J.: On the relationship between Arctic ice clouds and polluted air masses over the North Slope of Alaska in April 2008, Atmos. Chem. Phys., 14, 1205-1224, doi:10.5194/acp-14-1205-2014, 2014.

Legrand, M., C. Pietras, G. Brogniez, M. Haeffelin, N. Abuhassan, and M. Sicard, 2000: A High-Accuracy Multiwavelength Radiometer for In Situ Measurements in the Thermal Infrared. Part I: Characterization of the Instrument. J. Atmos. Oceanic Technol., 17, 1203–1214, doi: 10.1175/1520-0426(2000)017<1203:AHAMRF>2.0.CO;2.

Libois, Q., Ivanescu, L., Blanchet, J.-P., Schulz, H., Bozem, H., Leaitch, W. R., Burkart, J., Abbatt, J. P. D., Herber, A. B., Aliabadi, A. A., and Girard, É.: Airborne observations of far-infrared upwelling radiance in the Arctic, Atmos. Chem. Phys., 16, 15689-15707, doi:10.5194/acp-16-15689-2016, 2016.

Mariani, Z., Strong, K., Wolff, M., Rowe, P., Walden, V., Fogal, P. F., Duck, T., Lesins, G., Turner, D. S., Cox, C., Eloranta, E., Drummond, J. R., Roy, C., Turner, D. D., Hudak, D., and Lindenmaier, I. A.: Infrared measurements in the Arctic using two Atmospheric Emitted Radiance Interferometers, Atmos. Meas. Tech., 5, 329-344, doi:10.5194/amt-5-329-2012, 2012.

Nakajima, T. and M. King, 1990: Determination of the Optical Thickness and Effective Particle Radius of Clouds from Reflected Solar Radiation Measurements. Part I: Theory. J. Atmos. Sci., 47, 1878–1893, doi: 10.1175/1520-0469(1990)047<1878:DOTOTA>2.0.CO;2.

Rowe, P., L. Miloshevich, D. Turner, and V. Walden, 2008: Dry Bias in Vaisala RS90 Radiosonde Humidity Profiles over Antarctica. J. Atmos. Oceanic Technol., 25, 1529–1541, doi: 10.1175/2008JTECHA1009.1.

Sassen, K., Z. Wang, and D. Liu (2008), Global distribution of cirrus clouds from CloudSat/Cloud-Aerosol Lidar andInfrared Pathfinder Satellite Observations (CALIPSO) measurements, J. Geophys. Res., 113, D00A12, doi:10.1029/2008JD009972

Sourdeval, O., -Labonnote, L. C., Brogniez, G., Jourdan, O., Pelon, J., and Garnier, A.: A variational approach for retrieving ice cloud properties from infrared measurements: application in the context of two IIR validation campaigns, Atmos. Chem. Phys., 13, 8229-8244, doi:10.5194/acp-13-8229-2013, 2013.

Stephens, G., 2005: Cloud Feedbacks in the Climate System: A Critical Review. J. Climate, 18, 237–273, doi: 10.1175/JCLI-3243.1.

Treffeisen, R., Krejci, R., Ström, J., Engvall, A. C., Herber, A., and Thomason, L.: Humidity observations in the Arctic troposphere over Ny-Ålesund, Svalbard based on 15 years of radiosonde data, Atmos. Chem. Phys., 7, 2721-2732, doi:10.5194/acp-7-2721-2007, 2007.

Turner, D.D., B.M. Lesht, S.A. Clough, J.C. Liljegren, H.E. Revercomb, and D.C. Tobin, 2003: Dry bias and variability in Vaisala radiosondes: The ARM experience. J. Atmos. Oceanic Technol., 20, 117-132.

Waliser, D., et al. (2009), Cloud ice: A climate model challenge with signs and expectations of progress, J. Geophys. Res.,114, D00A21, doi:10.1029/2008JD010015.

Wang, J., L. Zhang, A. Dai, F. Immler, M. Sommer, and H. Vömel, 2013: Radiation Dry Bias Correction of Vaisala RS92 Humidity Data and Its Impacts on Historical Radiosonde Data. J. Atmos. Oceanic Technol., 30, 197–214, doi: 10.1175/JTECH-D-12-00113.1.

Winker, D., J. Pelon, J. Coakley, S. Ackerman, R. Charlson, P. Colarco, P. Flamant, Q. Fu, R. Hoff, C. Kittaka, T. Kubar, H. Le Treut, M. McCormick, G. Mégie, L. Poole, K. Powell, C. Trepte, M. Vaughan, and B. Wielicki, 2010: The CALIPSO Mission: A Global 3D View of Aerosols and Clouds. Bull. Amer. Meteor. Soc., 91, 1211–1229, doi: 10.1175/2010BAMS3009.1.

---

## Author Comment (AC2) · 12 Mar 2017

Response to D. P. Donovan (Referee #2)

The reviewer's comments are in black and our answers are in blue. Snippets of text from the submitted manuscript are in italics while modifications of the manuscript are shown in bold italics. The pages and lines reported here correspond to the submitted manuscript.

This paper describes a novel method to determine (thin) ice cloud optical depth as well as some limited mean particle size information using a combination of lidar data, IR radiometry and atmospheric thermodynamic state information. The technique appears robust and appears reasonably easy to implement. It could also be a candidate for network deployment.
The paper is, in the main, clear and well-written and worthy of publication. Another reviewer has already noted a number of issues. I have noted a further few aspects that should be improved before final acceptance.

We would like to thank Dave Donovan for his pertinent and informative comments. We provide below a point-by-point reply to his comments.

P1: line 1: What type of profile information? Please be specific here.

We have completed the missing information. P1 L1: *Multi-band downwelling thermal measurements of zenith sky radiance, along with **cloud boundary heights** , ...*

P11: Lines 25-28: The authors should expand the lidar multiple-scattering discussion. It is not quite satisfying/convincing to me. I agree that the application of Eloranta's formalism is appropriate, (it would be useful if they specified the equation number of the formula they used) however, they must follow through to discuss the errors in terms of optical depth and not leave the discussion solely in terms of Pt/P1. Also, the discussion must be made much clearer.

For example, (looking at Eq. 4) it is clear that the important factor in determining the COD is the ratio of the signals at cloud base and cloud top. Thus, the relevant quantity for determining the effect of MS is the Pt/P1 ratio at cloud top (since at cloud-base Pt=0). However, curiously, the worst-case cloud-base height is defined as 5.5 km but the cloud top altitude is not specified.

Assuming that the worst-case Pt/P1 ratio value of 60% quoted by the authors is indeed the value at cloud-top. The error in COD induced by MS effects for the worst-case scenario can be easily calculated. Using Eq.4 it is easy to show that the effect of MS will be to lower the retrieved extinction by an amount given by
dCOD_ms = -0.5 log (1.0+Pt/P1)
and Pt=0.6 P1 implies that dCOD_ms= 0.23 which is about a -10% bias.

Moving on, it is not clear what the authors mean by the "overall average value for the Pt/P1 upper limit". Is that the altitude averaged value for the just described "worst-case" scenario or the average ratio at cloud-top over the investigedcases ?If the former, then it is not a useful quantity. If the latter then that would correspond to a COD retrieval bias of -0.05 which could be significant for many of the results presented in this paper (e.g. see Fig .6 before about 19:00). Indeed accounting for the MS effects may bring the IR radiometer and lidar results more into line with each other values.

The authors have enough information to define the COD, cloud geometry and particle size ranges they are dealing with. Using this information and Eloranta's model I think with not too much effort they

could define and apply a mean eta factor as was done in the paper by Platt which they reference.

Multiple-scattering (MS) is an important question and we agree that this part could be improved. The former approach (using the formula (16) in Eloranta (1998)) has been improved by using his model for all the cases (Figure 1). As Dr. Donovan correctly wrote, the MS is maximum at cloud top height (or at least, very close to top height depending on the backscatter profile) and we changed it in the text. We also pursued this study to estimate the MS coefficient, as defined in Platt (1973), to take account of the MS effects. We hope that the additional statements will help the readers to understand the real but limited (less than 10 %) impact of MS in the present cloud study.

*Due to a very small angular field-of-view of the AHSRL receiver (45 μrad) it is common to assume the molecular backscatter cross section is not affected by multiple scattered photons (Eloranta et al., 2007). However to better quantify the effect of multiple scattering, we applied a practical model for the calculation of multiply scattered lidar returns, developed by Eloranta (1998) to the 150 cloud cases (not shown here). The inputs included molecular and particular backscatters, effective diameters (inferred from AHSRL+MMCR technique) and cloud height boundaries. The practical model computed the signal for all orders of multiple scattering but we were limited to the 4th order due to computation time. The impact of MS can then be evaluated in term of COD, as the multiple scattering lowers the retrieved extinction. From Equation (4), we define ΔCOD_ms as :*
*ΔCOD_ms = -0.5 log (1.0+Pt/P1)*
*where Pt/P1 is the signal for all orders of multiple scattering over the single scattering return, and is an output of the MS model. Lastly, this term can be used to correct the retrieved optical depth by inserting a multiplicative factor η to model the reduction of the extinction coefficient, as used in Platt (1973). The parameter η is not constant (Bissonnette, 2005) and Platt argued that it should vary between 0.5 and 1. A linear fit would lead to the conclusion that in this present study, η equals 0.95. From here, this factor has been used to correct COD retrieved by lidar in the rest of the manuscript.*

[Figure]

Figure 1: Difference in COD due to multiple scattering, according to the calculations of Eloranta's model (1998). The black squares represent the 150 cases selected in this study.

Page 20: Line 5-10: Can the authors please comment on how realistic it is to assume that all the measurement errors on the IR BTs are indeed independent?

From an error analysis of a very similar instrument (we used a newer version with 2 additional channels), in Legrand et al. (2000), we cannot state that the measurement errors are wavelength-independent. On the contrary, it seems that the global uncertainties for each channel depend on the detector temperature (Legrand et al, 2000; Brogniez et al., 2003).
We understand that our choice of assuming uncorrelated noise is simplistic but this assumption is done in numerous studies (e.g. Daniel et al., 2003; Turner, 2003; Sourdeval et al., 2013; Köhler et al., 2015; Sourdeval et al., 2016). We assume that the correlated errors would be reduced when the cloud dominates the measured signal (as the measured Tb is strongly linked to COD).
Some corrections can also be done to reduce noise level and therefore to minimize the off-diagonal elements of the covariance matrix. For example, a noise filter using principal component analysis was applied to P-AERI data in this study to reduce uncorrelated error (Turner et al., 2006).

Page 20: Line 14: sigma/sqrt(1000). What is the significance of the sqrtterm ?

The goal of applying the sqrt term was to have the same order of magnitude in the total error covariance matrix. By applying this term, we ensure to keep the spectral variation of the forward model errors.

References:

Brogniez, G., C. Pietras, M. Legrand, P. Dubuisson, and M. Haeffelin, 2003: A High-Accuracy Multiwavelength Radiometer for In Situ Measurements in the Thermal Infrared. Part II: Behavior in Field Experiments. J. Atmos. Oceanic Technol., 20, 1023–1033, doi: 10.1175/1520-0426(2003)20<1023:AHMRFI>2.0.CO;2.

Bissonnette, L. R., 2005: Lidar and multiple scattering.LidarRange-Resolved Optical Remote Sensing of the Atmosphere,C. Weitkamp, Ed., Springer, 43–104

Daniel, J. S., S. Solomon, H. L. Miller, A. O. Langford, R. W. Portmann, and C. S. Eubank, Retrieving cloud informationfrom passive measurements of solar radiation absorbed by molecular oxygen and O2-O2, J. Geophys. Res., 108(D16), 4515,doi:10.1029/2002JD002994, 2003.

Eloranta, E. W, Practical model for the calculation of multiply scattered lidar returns, Appl. Opt. 37, 2464-2472, 1998

Köhler, P., Guanter, L., and Joiner, J.: A linear method for the retrieval of sun-induced chlorophyll fluorescence from GOME-2 and SCIAMACHY data, Atmos. Meas. Tech., 8, 2589-2608, doi:10.5194/amt-8-2589-2015, 2015.

Legrand, M., C. Pietras, G. Brogniez, M. Haeffelin, N. Abuhassan, and M. Sicard, 2000: A High-Accuracy Multiwavelength Radiometer for In Situ Measurements in the Thermal Infrared. Part I: Characterization of the Instrument. J. Atmos. Oceanic Technol., 17, 1203–1214, doi: 10.1175/1520-0426(2000)017<1203:AHAMRF>2.0.CO;2.

Platt, C., 1973: Lidar and Radiometric Observations of Cirrus Clouds. J. Atmos. Sci., 30, 1191–1204, doi: 10.1175/1520-0469(1973)030<1191:LAROOC>2.0.CO;2.

Sourdeval, O., Labonnote, L. C., Brogniez, G., Jourdan, O., Pelon, J., and Garnier, A.: A variational approach for retrieving ice cloud properties from infrared measurements: application in the context of two IIR validation campaigns, Atmospheric Chemistry and Physics,13, 8229–8244, doi:10.5194/acp-13-8229-2013, http://www.atmos-chem-phys.net/13/8229/2013/, 2013.

Sourdeval, O., C.-Labonnote, L., Baran, A. J. and Brogniez, G. (2015), A methodology for simultaneous retrieval of ice and liquid water cloud properties. Part I: Information content and case study. Q.J.R. Meteorol. Soc., 141: 870–882. doi:10.1002/qj.2405

Turner, D.D., 2003: Microphysical properties of single and mixed-phase Arctic clouds derived from ground-based AERI observations. Ph.D. Dissertation, University of Wisconsin - Madison, Madison, Wisconsin, 167 pp.

Turner, D. D., Knuteson, R. O., Revercomb, H. E., Lo, C., and Dedecker, R. G.: Noise Reduction of Atmospheric Emitted Radiance Interferometer (AERI) Observations Using Principal Component Analysis, J. Atmos. Oceanic Technol., 23, 1223–1238,doi:10.1175/JTECH1906.1, http://dx.doi.org/10.1175/JTECH1906.1, 2006.

---

## Author Comment (AC3) · 12 Mar 2017

Response to Anonymous Referee #3

The reviewer's comments are in black and our answers are in blue. Snippets of text from the submitted are in italics while modifications of the manuscript are shown in bold italics. The pages and lines reported here correspond to the submitted manuscript.

This paper reports on the development and testing of new LIRAD method that utilizes thermal infrared bands available on the CIMEL CE-312 radiometer, which is considerably cheaper than the AERI. The latter has been used in the recent past to provide the thermal components to retrieve COD and Deff from the surface. The implicit motivation for this study seems to be that the LIRAD method can be used more often if the CIMEL-312 could be used instead of the AERI, since it is attempting to do the same things that AERI already provides. The explicit motivation is that the ability to discriminate between small and large Deff values would help the study of aerosol/cloud interactions in polar regions, especially regarding aerosol influence on precipitative cooling and (as inferred from the DLCRF computations) on radiative heating of the surface. The writing is of average quality, but has some grammatical issues. The paper is scientifically sound in its approach, but the results and method do not appear to be particularly new.

We are grateful to this reviewer for the helpful comments. We provide below a point-by-point reply to his comments.

The algorithm is new in that it uses specific bands not previously employed in the AERI-LIRAD approach. However, they are not all that different from the AERI microwindows. Except for lacking the wavelengths longer than 14 μm, this is essentially the same algorithm. If not, then it needs more contrasting with the MIXCRA. It relies on the phase being known already, whereas, I believe, the longer wavelengths used in the MIXCRA were primarily for phase discrimination. It is not surprising then that the results are quite close to those from the MIXCRA. The botton line that I believe the authors should address is "how many wavelengths are actually needed to replace the 19 microwindows of MIXCRA having lambda < 13 μm to perform the retrieval?" Could you do it with 2 or 3 channels? There appears to be a lot of information redundancy in the bands that were used.

The question about information redundancy is pertinent. We believe the information in section 4 (the discussions centered on Figures 2, 3 and 4) shows the sensitivity of each band to the key radiative transfer parameters and therefore helps to demonstrate the importance of those bands. The $T_b$ separation of the spectral curves in Figures 3a and 3b is a progressively damped out sensitivity to the cold-space temperature that is essentially controlled by water vapour absorption at COD = 0.The variation of the brightness temperature with $D_{eff}$ in Figure 3b is reflective of essentially two dominant optical mechanisms juxtaposed on the progressive warming of the cold background : (i) the classical increase in the extinction efficiency with increasing $D_{eff}$ (and, optically speaking, with decreasing wavelength) in what we call the small-particle Angstrom-exponent regionand (ii) the filtering / masking of this robust monoticity (linearity on a log-plot) caused by significant variations in the real and complex parts of the refractive index in the last 3 bands (notably the last 2 bands). This translates, in a $1^{st}$ order sense, to an Angstrom type of slope requirement for at least 2 short wavelength bands (3 for better redundancy).The monotonic dependence of brightness temperature as a function of COD in Figure 3a is also dependant on the interplay of the 2 extinction efficiency influences and the progressive warming of the cold-space temperature. It is clear that any single band would fare quite well in a $T_b$ inversion to extract COD but that the most transparent bands to water vapour would be more sensitive (the 11.3 μm band in Figure 3a). As the brightness temperature of all bands is also sensitive to a variety of parameters (Figure 4), it is important, we believe, to maintain a certain redundancy in order to ensure a

robust retrieval.

To further support this answer to the reviewer's question, we performed retrievals with different band configurations and compared retrieval statistics for the COD and $D_{eff}$ retrievals (tables below). The order of the elimination of these bands was based on a progressive increase in per-band information content (roughly from minimum to maximum $dX / dTb_i$ where $X = COD$ or $D_{eff}$ and where $Tb_i$ is the brightness temperature for band i in the region of greatest sensitivity to X)

Table 1: Retrieval statistics for different band configurations.

| Bands used (µm) | COD retrieval ($R^2$– RMS Errors) | Deff retrieval (overall accuracy) |
| --- | --- | --- |
| **8.4, 8.7, 9.2, 10.7, 11.3, 12.7** | **0.95 – 0.09** | **83%** |
| 8.7, 9.2, 10.7, 11.3, 12.7 | 0.95 – 0.09 | 75% |
| 9.2, 10.7, 11.3, 12.7 | 0.95 – 0.09 | 72% |
| 10.7, 11.3, 12.7 | 0.95 – 0.09 | 71% |
| 10.7, 11.3 | 0.95 – 0.09 | 74% |
| 11.3 | 0.95 – 0.09 | 69% |

| Bands used (µm) | COD retrieval ($R^2$ – RMS Errors) | Deff retrieval (overall accuracy) |
| --- | --- | --- |
| **8.4, 8.7, 9.2, 10.7, 11.3, 12.7** | **0.95 – 0.09** | **83%** |
| 8.4, 8.7, 9.2, 10.7, 11.3 | 0.95 – 0.09 | 81% |
| 8.4, 8.7, 9.2, 10.7 | 0.95 – 0.09 | 77% |
| 8.7, 9.2, 10.7 | 0.95 – 0.10 | 73% |
| 8.7, 9.2 | 0.94 – 0.11 | 67% |
| 9.2 | 0.94 – 0.11 | 67% |

The COD retrieval statistics are similar even if only one band is used in the retrieval algorithm (where that one band is the most $T_b$ sensitive, 11.3 µm band). In the case of the $D_{eff}$ retrieval, the overall accuracy decreases slowly until the Angstrom slope information is eliminated by reducing the band number from 2 to 1.

The validation effort includes the comparisons with MIXCRA (noted above) and with lidar for COD. This begs the question: If the lidar is required for the LIRAD method and it produces a reliable COD (it is used as a reference), why then is the IR method used to estimate COD? Why not simply estimate $D_{eff}$ using the IR data and the lidar COD as input? Lidar retrievals of COD are quite common for most lidars deployed for surface observations. What am I missing? How will phase be determined if this approach is implemented elsewhere?

The radiometer is designed to be a portable instrument and can be easily deployed to a remote station. We could certainly use the lidar COD as input but the requirementforapplying our retrieval method is only cloud base altitude information (and ideally also cloud thickness). This could be obtained from a ceilometer or a portable, low-power lidar (CE370 LiDAR from CIMEL, for example). Put another way, the advanced capabilities of a lidar such as the AHSRL were needed for the validation but only a ceilometer was required to estimate the required input parameters to our passive retrieval algorithm (the

AHSRL is overkill in the latter case).

We inserted to following sentences in the conclusion of the paper: P.20 L.12 *One of the perspectives could be to deploy this technique as part of a network of low-cost and robust instruments to monitor arctic clouds. Because their occurrence, type and altitude are spatially inhomogeneous (according to Eastman and Warren (2010) and Shupe et al. (2011)), we believe that additional ground-based stations would be helpful to broaden our knowledge of arctic ice clouds.*

Some authors have proposed to use the absorption coefficient differences between 10 and 13μm for phase discrimination (ice being absorbing than water;see, for example, Baum et al., 2000). However,inasmuchas absorption is also a function of particle size (see Fig. 2), it could be difficult, to separate ice and water in the case of small particles (Turner, 2003). This is why the bands between 16 and 20 μm are used in the case of MIXCRA.
Moreover, as shown by Shupe (2011), the frequency of liquid clouds (either liquid-only ormixed-phase clouds) is low at Eureka (annual average of 30%, mainly occurring in the summer and autumn and below 20% during the wintertime, when the cases in this study were chosen).

I think this paper can be published, but it needs some major revisions to provide better justification as to why it is necessary and to flesh out the analysis by addressing the questions above.

Minor comments
P1, L24: "temperatures" does not belong here

It has been removed.

P3, L4: " aerosols acting as ice and water cloud nuclei, cloud microphysics, precipitation and radiation" does not make any sense. Please reorder this so that aerosols do not appear to act as cloud microphysics.

This sentence was rewritten: P3 L4 : *Much of the recent research has been focused on aerosol-cloud interactive processes involving aerosols acting as ice and water cloud nuclei **and their subsequent affect on** cloud microphysics, precipitation and radiation.*

P3, L14: You may want to modify the sentence with "could possibly lead to" or something similar, since the "dehydration greenhouse feedback" is only a proposed mechanism.

This sentence was rewritten as suggested: P3 L14 : *In terms of the purpose and motivation for this paper, we note that the presence of sulphuric-acid bearing aerosols (viz., Arctic haze) can significantly increase the size of ice particles (relative to the size of ice particles formed from more pristine, low acid aerosols or supercooled droplets). This process can **cause**  enhanced precipitation and important cooling effects during the polar winter **and could lead to a dehydration greenhouse feedback (DGF) effect, as proposed by Blanchet and Girard (1994).***

P4, L3: Please "COD" for singular and "CODs" or "COD values" for plural here and throughout the paper

The consistency of the use of the singular and plural forms was corrected in the paper.

P4, L9: "northern most" should be one word

Corrected.

P4, L14: if the same summary is given by both references, then leave one out. If two different summaries are provided, then change everything to the plural form.

We chose to keep the reference of Bourdages et al. (2009) as the technical specifications were more detailed.

P4, L17: "in the order of" should be "on the order of"

Corrected.

P5, L24-25: The <2% refers only to downwelling radiation viewed at the zenith. It can be up to10% for other viewing conditions, particularly for upwelling radiation (e.g., Minnis et al., JAS 93). The viewing limitation should be highlighted again when referring to the 2%.

We have added this correction: P5 L24: *Platt (1973) (and later authors such as **Turner and Lohnert, 2014, for wavelengths higher than 10 μm and shorter than 16 μm**) indicated that less than 2% of the **zenith-looking, downwelling** radiation emitted by a cloud was due to scattering.*

P7,L16: COD can only have an amplitude if you are referring to its oscillation with time or space. Otherwise, please refer to it as magnitude or value.

That is true, we were talking about its magnitude. P7, L16: *At COD **magnitudes**  greater than 2-3, ...*

P8, L3-6: Sentence should be broken up for clarity.

The rewritten sentences are: P8 L3-L6: *The simulation results in Figure 4 detail the band dependent effects of six different parameters by comparing changes in Tb induced by each parameter individually. **These were obtained** using a random number generator with a normal probability distribution whose mean and standard deviation was controlled by the six parameter values of Table 2 (COD, $D_{eff}$, cloud base height, cloud thickness, column integrated water vapor of the atmosphere (WVC) and particle shape).*

P8, L16: cloud altitude and thickness uncertainties do not have amplitudes in this context. See comment above.

It was corrected. P8 L16: *If the altitude and the thickness of the clouds are known from vertical lidar (and radar) profiles then the **magnitude**  of the altitude and cloud thickness uncertainties of Figure ...*

P10, L1: Here and elsewhere, the form of the modifier does not need to match that of the noun. Should be "ice cloud retrievals".

Corrected.

P11, L10: "mean't" appears to be a typo.

Corrected.

P12, L12: "produces gives", please use one or the other

We removed "produces".

P12, L18: should be "hydrometeor diameter"

Corrected.

P16, L4-5: "ommission" has only one "m"

Corrected.

References:
Baum, B. A., P. F. Soulen, K. I. Strabala, M. D. King, S. A. Ackerman, W. P. Menzel, and P. Yang, 2000: Remote sensing of cloud properties using MODIS airborne simulator imagery during SUCCESS: 2. Cloud thermodynamic phase, J. Geophys. Res., 105(D9), 11781–11792, doi:10.1029/1999JD901090.

Eastman, Ryan, Stephen G. Warren, 2010: Arctic cloud changes from surface and satellite observations. J. Climate, 23, 4233–4242.doi: 10.1175/2010JCLI3544.1

Shupe, M.D., 2011: Clouds at Arctic Atmospheric Observatories, Part II: Thermodynamic phase characteristics. J. Appl. Meteor. Clim., 50, 645-661.

Turner, D.D., 2003: Microphysical properties of single and mixed-phase Arctic clouds derived from ground-based AERI observations. Ph.D. Dissertation, University of Wisconsin - Madison, Madison, Wisconsin, 167 pp. Available from http://www.ssec.wisc.edu/library/turnerdissertation.pdf

Turner, D. D. and U. Löhnert, 2014: Information Content and Uncertainties in Thermodynamic Profiles and Liquid Cloud Properties Retrieved from the Ground-Based Atmospheric Emitted Radiance Interferometer (AERI). J. Appl. Meteor. Climatol., 53, 752–771, doi: 10.1175/JAMC-D-13-0126.1